# Dysregulation of mTOR signaling mediates common neurite and migration defects in both idiopathic and 16p11.2 deletion autism neural precursor cells

**Smrithi Prem[1,2]\*, Bharati Dev[1], Cynthia Peng[1], Monal Mehta[2,3], Rohan Alibutud[4], Robert J Connacher[1,2], Madeline St Thomas[1,2], Xiaofeng Zhou[1], Paul Matteson[1,3], Jinchuan Xing[4], James H Millonig[1,3]\*†, Emanuel DiCicco-Bloom[1,5]\*†**

[1]Department of Neuroscience and Cell Biology, Rutgers Robert Wood Johnson Medical School, Piscataway, United States; [2]Graduate Program in Neuroscience, Rutgers University, Piscataway, United States; [3]Center for Advanced Biotechnology and Medicine, Rutgers University, Piscataway, United States; [4]Department of Genetics, Rutgers University, Piscataway, United States; [5]Department of Pediatrics, Rutgers Robert Wood Johnson Medical School, New Brunswick, United States

**\*For correspondence:**
prems@pennmedicine.upenn.
edu (SP);
millonig@cabm.rutgers.edu
(JHM);
diciccem@rwjms.rutgers.edu
(EDiC-B)

†These authors contributed
equally to this work

**Competing interest:** The authors
declare that no competing
interests exist.

**Reviewing Editor:** Genevieve
Konopka, University of Texas
Southwestern Medical Center,
United States

**Abstract** Autism spectrum disorder (ASD) is defined by common behavioral characteristics, raising the possibility of shared pathogenic mechanisms. Yet, vast clinical and etiological heterogeneity suggests personalized phenotypes. Surprisingly, our iPSC studies find that six individuals from two distinct ASD subtypes, idiopathic and 16p11.2 deletion, have common reductions in neural precursor cell (NPC) neurite outgrowth and migration even though whole genome sequencing demonstrates no genetic overlap between the datasets. To identify signaling differences that may contribute to these developmental defects, an unbiased phospho-(p)-proteome screen was performed. Surprisingly despite the genetic heterogeneity, hundreds of shared p-peptides were identified between autism subtypes including the mTOR pathway. mTOR signaling alterations were confirmed in all NPCs across both ASD subtypes, and mTOR modulation rescued ASD phenotypes and reproduced autism NPC-associated phenotypes in control NPCs. Thus, our studies demonstrate that genetically distinct ASD subtypes have common defects in neurite outgrowth and migration which are driven by the shared pathogenic mechanism of mTOR signaling dysregulation.

## Editor's evaluation

This important study identifies common developmental defects in neural precursor cells derived from idiopathic and 16p11.2 deletion ASD individuals. Convincing evidence is presented that mTOR signaling defects are a shared pathogenic mechanism underlying the cellular defects in these genetically distinct ASD subtypes. This work will be of interest to researchers studying neurodevelopment and related disorders.

## Introduction

Autism spectrum disorders (ASD) are characterized by deficits in social interaction and communication and the presence of repetitive and restrictive interests and behaviors. Though these common impairments define ASD, there is marked heterogeneity in both clinical presentation and etiological underpinnings among affected individuals (*Jones and Klin, 2009*; *Kim and Lord, 2013*, *Betancur,*

**eLife digest** Although the clinical presentation of individuals with autism spectrum disorder (ASD) can vary widely, the core features are repetitive behaviors and difficulties with social interactions and communication. In most cases, the cause of autism is unknown. However, in some cases, such as a form of ASD known as 16p11.2 deletion syndrome, specific genetic changes are responsible.

Despite this variability in possible causes and clinical manifestations, the similarity of the core behavioral symptoms across different forms of the disorder indicates that there could be a shared biological mechanism. Furthermore, genetic studies suggest that abnormalities in early fetal brain development could be a crucial underlying cause of ASD. In order to form the complex structure of the brain, fetal brain cells must migrate and start growing extensions that ultimately become key structures of neurons.

To test for shared biological mechanisms, Prem et al. reprogrammed blood cells from people with either 16p11.2 deletion syndrome or ASD with an unknown cause to become fetal-like brain cells. Experiments showed that both migration of the cells and their growth of extensions were similarly disrupted in the cells derived from both groups of individuals with autism.

These crucial developmental changes were driven by alterations to an important signaling molecule in a pathway involved in brain function, known as the mTOR pathway. However, in some cells the pathway was overactive, whereas in others it was underactive. To probe the potential of the mTOR pathway as a therapeutic target, Prem et al. tested drugs that manipulate the pathway, finding that they could successfully reverse the defects in cells derived from people with both types of ASD.

The discovery that a shared biological process may underpin different forms of ASD is important for understanding the early brain changes that are involved. A common target, like the mTOR pathway, could offer hope for treatments for a wide range of ASDs. However, to translate these benefits to the clinic, further research is needed to understand whether a treatment that is effective in fetal cells would also benefit people with autism.

*2011*). In turn, this heterogeneity raises the possibility that there are 'different' ASD subtypes (*Ousley and Cermak, 2014*; *Eaves et al., 1994*; *Beglinger and Smith, 2001*). Yet, despite this heterogeneity, all autism subtypes are united by their core behavioral phenotypes. Consequently, we postulate that there may be common pathways linking different autism subtypes. Yet, few studies have compared multiple ASD subtypes to investigate possible commonalities.

About 10–15% of autism cases are genetically defined and caused by single gene mutations or copy number variants (CNVs) while most of ASD etiology remains undefined or idiopathic. Yet, much of our knowledge of ASD pathophysiology is derived from rodent studies of syndromic rare variant genes. Consequently, our insight into a majority of ASD pathophysiology has been limited and we have little knowledge of the similarities and differences between idiopathic and genetically defined autisms. Several studies have uncovered the convergence of both rare and common variant ASD risk genes onto the developing mid-fetal cerebral cortex (8–24 weeks) (*Willsey et al., 2013*; *Parikshak et al., 2013*; *Satterstrom et al., 2020*; *Grove et al., 2019*). During this period, neural precursor cells (NPCs) undergo proliferation, migration, and differentiation to form neurons and establish normal brain cytoarchitecture (*Liszewska and Jaworski, 2018*). Now, with the advent of induced pluripotent stem cell (iPSC) technology, we can study human neural development in cells derived from individuals with both genetic and idiopathic neuropsychiatric disorders. iPSC studies of monogenic, CNV, and idiopathic ASD have observed alterations in basic developmental processes, such as synapse formation (*Acab and Muotri, 2015*; *Yang and Shcheglovitov, 2020*). However, many of these studies focused on terminally differentiated neurons, with very few examining developmental events that correspond to mid-fetal cortical development that precede synapse formation, such as neurite outgrowth and cell migration (*Packer, 2016*, *Prem et al., 2020*). Moreover, there is little insight into whether and how extracellular factors (EFs), which coordinate neurodevelopment by regulating signaling, contribute to ASD pathogenesis in human NPCs.

To begin addressing these questions, we studied early neurite development and cell migration using human iPSC-derived NPCs from two different autism subtypes – an idiopathic cohort (I-ASD) and a genetically defined CNV 16p11.2 deletion (16pDel) cohort. We demonstrate remarkable convergence

in neurite and migration phenotypes despite whole genome sequencing (WGS) showing no genetic overlap between the datasets. Specifically, we observed reductions in both neurite outgrowth and migration in NPCs derived from all individuals with ASD. Furthermore, to expand our understanding of NPC response to mid-fetal environmental stimuli, we stimulated NPCs with extracellular factors (EFs: growth factors, neuropeptides, neurotransmitters) that are expressed in the mid-fetal period and are known to regulate neurodevelopment. Surprisingly, our EF studies revealed that all unaffected NPCs responded to selected EFs with increases in neurite outgrowth and migration while I-ASD NPCs failed to respond to any of the tested EFs, suggesting the possibility of impaired signaling pathways. Thus, to identify mediating signaling pathways, an unbiased p-proteome analysis was conducted, which showed significant overlap in p-peptides between I-ASD and 16pDel datasets with less overlap in the total proteome. Bioinformatics revealed the mTOR pathway as one of the strongest points of convergence between the two ASD subtypes, which was supported by western analyses. Moreover, modulation of the mTOR pathway with small molecules rescued and reproduced ASD NPC phenotypes indicating that mTOR signaling defects are mechanistically linked to the neurodevelopmental phenotypes. Nonetheless, there were also ASD subtypes as evidenced by differential responses to EFs between I-ASD and 16pDel NPCs as well as distinct mTOR alterations including elevations and reductions. Overall, our studies indicate that despite the genetic and phenotypic heterogeneity, two distinct ASD subtypes have dysregulation of mTOR signaling which mediates the surprisingly uniform neurodevelopmental phenotypes.

## Results

### Patient cohort, rigor and reproducibility, and study design

The I-ASD cohort was selected from the previously reported New Jersey Language and Autism Genetics Study. From three separate families, iPSCs were generated from three male probands with ASD and their sex-matched unaffected siblings (Sib) served as controls (see Materials and methods).

For comparison, we selected a genetically defined ASD cohort with 16p11.2 deletion CNV. The CNV deletes 28 genes, increases autism risk by ~20-fold, and is one of the most common ASD-associated mutations (*Weiss et al., 2008*; *Niarchou et al., 2019*; *Qureshi et al., 2014*). This cohort consists of three individuals, two males (16pDel M-1, 16pDel M-2) and one female (16pDel F). For sex-matched controls, we selected three iPSC lines from the NIH Regenerative Medicine Program (NIH).

We analyzed these 12 individuals (six I-ASD-Sib and six 16pDel-NIH) extensively for developmental phenotypes utilizing two to five clones per individual (except for NIH) and two to five independent neural inductions to ensure extensive rigor and reproducibility (see Materials and methods and *Supplementary file 1*).

### Whole genome sequencing

WGS was conducted on I-ASD dataset and demonstrated that no individuals harbored the 16p11.2 deletion (*Supplementary file 2*). Moreover, no I-ASD individual had rare non-synonymous protein-coding variants in the 28 genes deleted in 16p11.2. Finally, there are no rare protein-coding non-synonymous genetic variants shared between all three I-ASD individuals. These data are consistent with the two datasets being genetically distinct with no common or shared genetic drivers.

### I-ASD and 16pDel have common defects in neurite outgrowth and cell migration

During the mid-fetal developmental window, there are a few critical cellular processes that occur and are essential for the normal function and architecture of the brain. These processes include proliferation, migration, and differentiation. Within differentiation, neurite outgrowth is a critical cytoskeleton-dependent process necessary for normal neural connectivity in the brain. Elaboration of neurites is an important mechanism to assess for proper development in neuronal cells and multiple studies of NDDs have reported impairment in neurite outgrowth (*Gleeson and Walsh, 2000*; *Uppal and Hof, 2013*; *Wegiel et al., 2010*; *Kathuria et al., 2018*; *Doers et al., 2014*; *Marchetto et al., 2010*; *Krey et al., 2013*; *Deshpande et al., 2017*; *Prem et al., 2020*). Thus, we first assessed neurite outgrowth in I-ASD NPCs at 48 hr (*Figure 1A and B*). Given the genetic heterogeneity of the I-ASD cohort, we expected to find personalized phenotypes. Yet, surprisingly, we found that all three I-ASD NPCs had

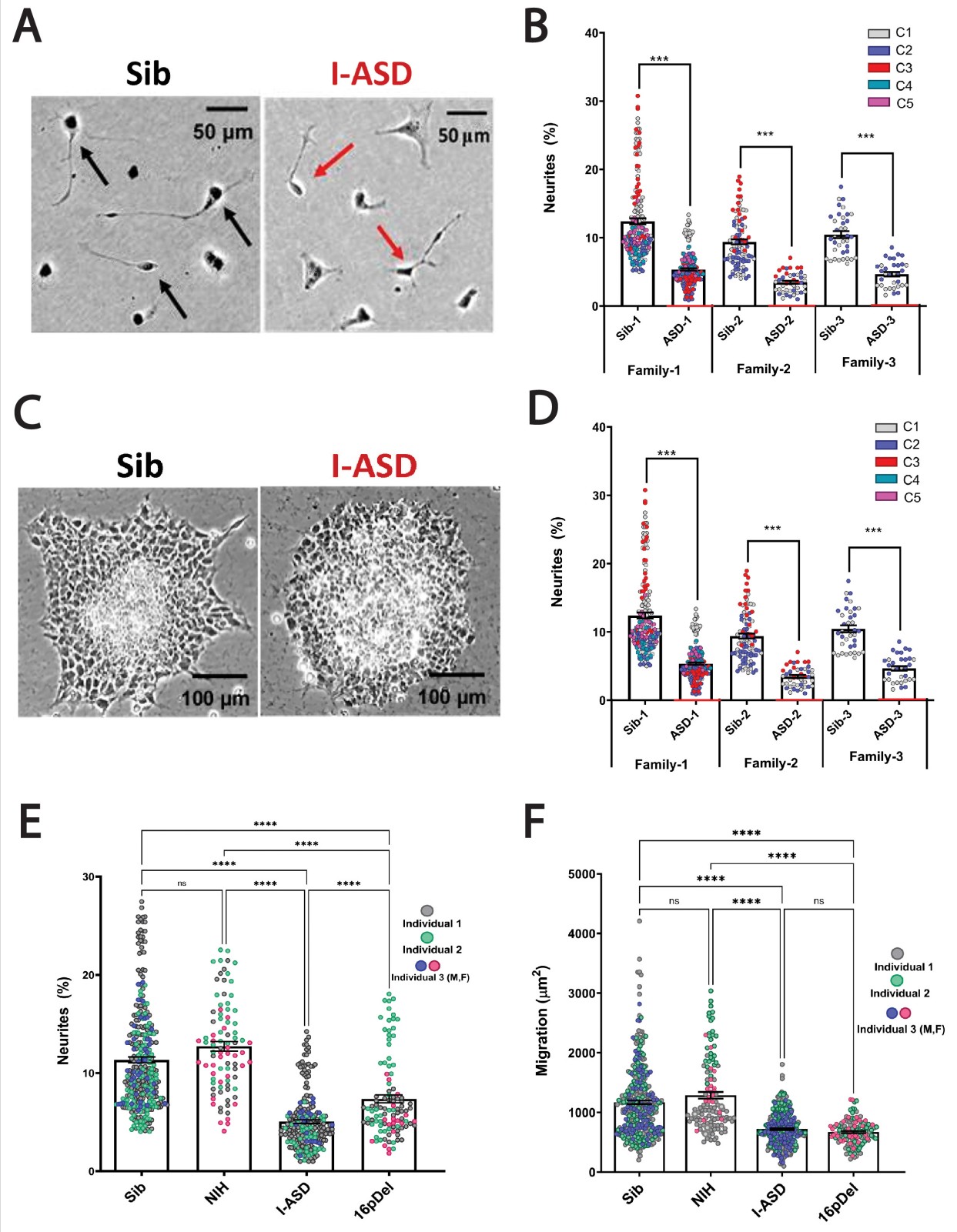

**Figure 1.** Reduced neurite outgrowth and cell migration in both I-ASD and 16pDel neural precursor cells (NPCs). (**A**) Representative image: neurite outgrowth at 48 hr: Sib NPCs have more cells with neurites (black arrows) than I-ASD NPCs (red arrows). (**B**) Quantification of % neurites in three pairs of I-ASD and Sibs: I-ASD NPCs have fewer neurites (%) than Sib NPCs in all families. For neurite graphs, each dot represents the % of cells with neurites in a single dish. Colors represent different clones. (**C**) Representative image: Sib neurospheres have migrating carpet of cells (dark) that move further

*Figure 1 continued on next page*

*Figure 1 continued*

than that of I-ASD neurosphere. (**D**) Cell migration in three pairs of I-ASD and Sibs: I-ASD NPCs migrate less than Sib NPCs in all families. For migration graphs, each dot represents an individual neurosphere. Student's t-test: all Sib vs I-ASD comparisons. (**E**) Neurite % in Sibs, NIH, I-ASD, and 16pDel ASD. Each dot represents data from a single dish with different colors denoting different individuals. Both 16pDel and I-ASD have reduced neurites compared to Sib and NIH NPCs. (**F**) NPC migration in Sibs, NIH, I-ASD, and 16pDel ASD. Both 16pDel and I-ASD NPCs have reduced migration compared to both Sib and NIH. One-way ANOVA for all 16pDel experiments. For all graphs: NS = p>0.05, p≤0.05 = *, p≤0.01 = **, p≤0.001 = ***, p≤0.0001 = ****. Error bars represent standard error of means (SEM). For detailed N values please see ***Supplementary file 1***.

The online version of this article includes the following figure supplement(s) for figure 1:

**Figure supplement 1.** Evaluation of the relationship between initial sphere size (ISS) and neurosphere migration.

a significantly lower percentage of neurites than their respective Sibs (***Figure 1B***, 50–60% reduction, p<0.001 for all families). To understand how I-ASD NPCs compared to genetically unrelated neurotypical individuals, we also compared I-ASD NPCs to Sibs from other families (e.g. I-ASD-1 to Sib-2). On average, all I-ASD NPCs had reduced neurites compared to all Sib NPCs.

Given the common neurite impairment in all I-ASD NPCs, we then extended our studies to another critical cytoskeleton-dependent mid-fetal developmental process, NPC migration, which occurs concurrently with neurite outgrowth and is similarly regulated during development (***Prem et al., 2020***). Moreover, alterations in migration have been reported in multiple NDDs as well (***Wegiel et al., 2010***; ***Wiegreffe et al., 2015***; ***Hori and Hoshino, 2017***; ***Prem et al., 2020***; ***Brennand et al., 2011b***; ***Brennand and Gage, 2011a***, ***Brennand et al., 2011c***, ***Brennand et al., 2015***; ***Pan et al., 2019***). For these studies, we plated neurospheres and at 48 hr measured the area covered by migrating NPCs (***Figure 1C***). Remarkably, we found that all three I-ASD patients exhibited decreased migration (~30–50%) compared to their Sib (p<0.001; ***Figure 1D***) as well as unrelated Sibs. Moreover, differences in migration area were not related to the initial neurosphere size (see Materials and methods and ***Figure 1—figure supplement 1***). Thus, despite genetic heterogeneity both neurite outgrowth and cell migration are similarly impaired in all three I-ASD NPCs.

This remarkable convergence prompted us to examine the 16pDel ASD cohort for these phenotypes. Fascinatingly, 16pDel NPCs have a lower percentage of neurites (7.4%) than both the Sib (11.3%, p<0.001) and NIH controls (12.7%, p<0.001) (***Figure 1E***). This indicates that defects in neurite outgrowth are a common feature across two genetically distinct subtypes of autism. Interestingly, 16pDel NPCs do have a higher percent of neurites than I-ASD (7.4% vs 5.1%, p<0.01) indicating potential subgroup differences. There was no difference in neurite extension between Sib NPCs (11.3%) and NIH NPCs (12.7%) (p=0.1).

Lastly, we assessed migration in 16pDel NPCs. Once again, like I-ASD NPCs, all three 16pDel NPCs have impaired migration when compared to Sib and NIH NPCs (***Figure 1F***). There were no differences in migration between 16pDel and I-ASD or Sib NPCs and NIH NPCs.

In summary, we have found that two genetically distinct ASD cohorts have surprisingly common dysregulation in two early neurodevelopmental processes, neurite outgrowth and cell migration. Given the marked heterogeneity of ASD, these results are striking and potentially suggest that common mechanisms regulate these neurodevelopmental defects.

## I-ASD and 16pDel NPCs have subtype-specific responses to EFs

Few iPSC studies have explored the roles of EFs in regulating human NPC development. We postulated that stimulating NPCs with EFs could reveal defects not apparent in control conditions and help identify underlying signaling defects. Thus, we stimulated our NPCs with nerve growth factor (NGF), PACAP, and 5-HT which regulate neurite outgrowth or cell migration in rodent studies (***Maisonpierre et al., 1990***; ***Hanswijk et al., 2020***; ***Dicicco-Bloom et al., 1998***; ***Lu and DiCicco-Bloom, 1997***). Furthermore, in prior studies in our lab, we have noted impaired response to EFs in rodent models of NDDs (***Rossman et al., 2014***).

We first tested EFs on the I-ASD cohort. Under PACAP, NGF, and 5-HT stimulation, Sib NPCs from all families had significant increases (45–75%, p<0.001) in neurites (***Figure 2A and B***, and ***Figure 2—figure supplement 1***). In contrast, all three I-ASD NPCs failed to respond to the three EFs (***Figure 2A and B***, and ***Figure 2—figure supplement 1***). The lack of response was not due to differential sensitivities or inappropriate doses of EF, as dose-response analyses indicate Sibs respond to PACAP and NGF

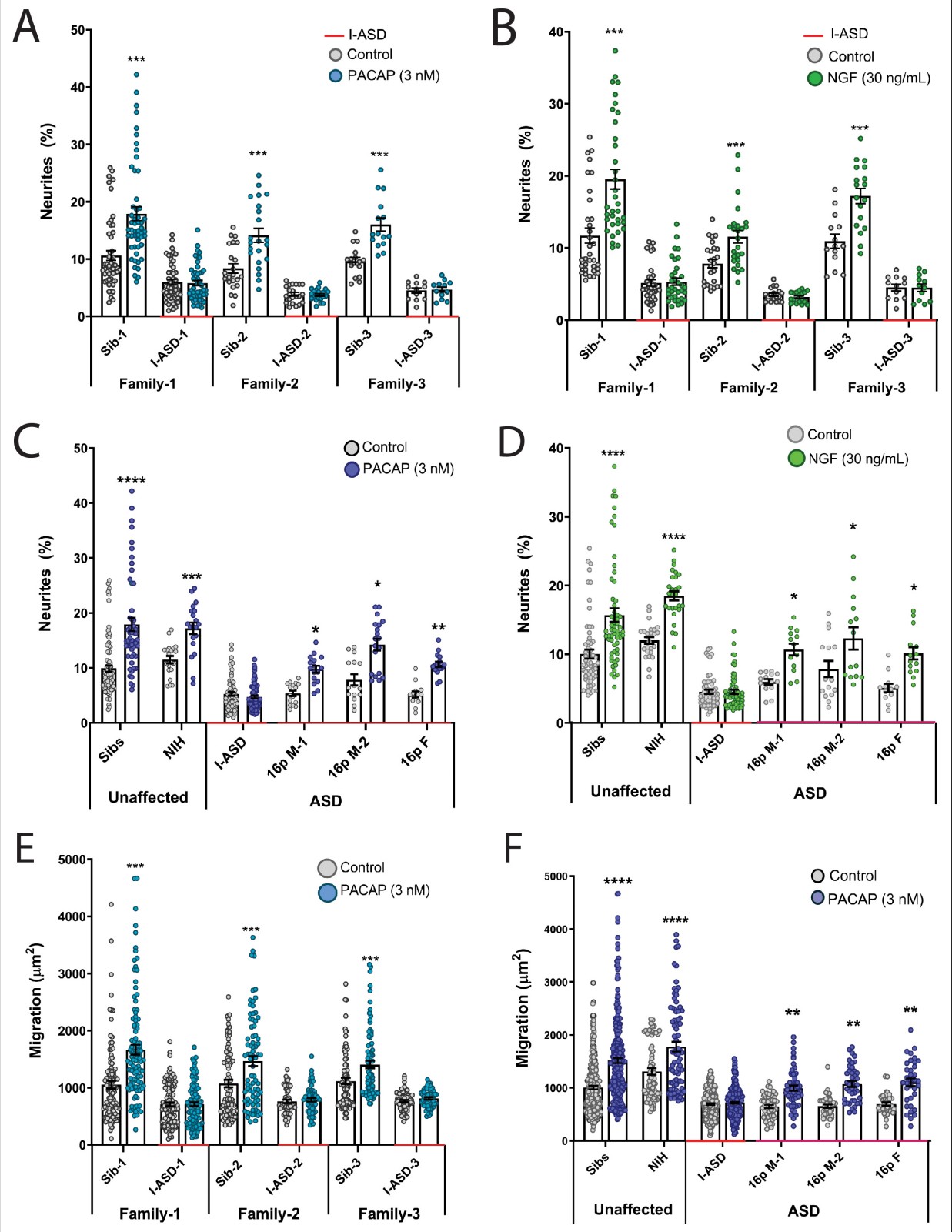

**Figure 2.** Autism spectrum disorder (ASD) subtype-specific extracellular factor (EF) responses in I-ASD and 16pDel neural precursor cells (NPCs). (**A and B**) 3 nM PACAP (**A**) and 30 ng/mL nerve growth factor (NGF) (**B**) increases neurite outgrowth in all Sibs but fails to stimulate neurite outgrowth in all I-ASD NPCs. (**C and D**) 3 nM PACAP (**C**) and 30 ng/mL NGF (**D**) stimulate neurite outgrowth in Sibs, NIH, and all three 16pDel NPC but not in I-ASD NPCs. (**E**) 3 nM PACAP increases cell migration in all Sib NPCs but fails to stimulate migration in all I-ASD NPCs. (**F**) 3 nM PACAP stimulates migration

*Figure 2 continued on next page*

*Figure 2 continued*

in Sibs, NIH, and all three 16pDel patient NPCs but does not change migration in I-ASD NPCs. Two-way ANOVA for all comparisons, p<0.01 for all unaffected vs affected comparisons. For all graphs: p≤0.05 = *, p≤0.01 = **, p≤0.001 = ***, p≤0.0001 = ****. Error bars represent SEM. For detailed N values please see *Supplementary file 1*.

The online version of this article includes the following figure supplement(s) for figure 2:

**Figure supplement 1.** Effects of 5-HT on I-ASD and Sib neural precursor cells (NPCs).

**Figure supplement 2.** Dose-response graphs for extracellular factors (EFs).

**Figure supplement 3.** Effects of 5-HT on 16pDel neural precursor cells (NPCs).

at multiple doses while ASD had no response at any physiological dose (*Figure 2—figure supplement 2*).

Next, we extended EF studies to the 16pDel cohort. Surprisingly, unlike I-ASD NPCs, all three 16pDel NPCs respond to EFs with increases in neurite outgrowth. As expected, like Sib controls, NIH NPCs also exhibit increases in neurite outgrowth by EF stimulation (*Figure 2C and D*, *Figure 2—figure supplement 3*).

We also applied EFs, specifically PACAP, a well-known regulator of migration in rodent models (*Falluel-Morel et al., 2005*), to our migration assay. Like our neurite results, increased cell migration in Sibs was observed but there was no effect of PACAP on migration in any of the three I-ASD NPCs (*Figure 2E*). Likewise, all three 16pDel NPCs and NIH NPCs respond to PACAP with increased migration (*Figure 2F*).

In summary, while our two ASD cohorts have common impairments in neurite outgrowth and migration, EF experiments uncovered subtype-specific response in I-ASD vs 16pDel. Thus, subtype-specific defects can be present alongside common neurobiological phenotypes.

## Phospho-proteomic analysis reveals dysregulated signaling in both ASD cohorts

Signaling pathways are central to the regulation of neurodevelopmental processes and signaling dysregulation has been implicated in NDD pathogenesis (*Lipton and Sahin, 2014*; *Takei and Nawa, 2014*; *Kelley et al., 2008*; *Pucilowska et al., 2012*; *Kwan et al., 2016*; *Wang et al., 2017*; *Waite and Eickholt, 2011*). Given the neurodevelopmental phenotypes present in all our ASD NPCs and the impaired EF responses in I-ASD, we postulated that dysregulated signaling pathways could be an underlying mechanism. Thus, we conducted unbiased proteomic and p-proteomic studies comparing Sib and I-ASD, Sib and 16pDel, and I-ASD and 16pDel.

We first analyzed the proteome and found surprisingly few changes in the I-ASD and 16pDel datasets relative to Sib. Initial data included over 9700 proteins, however, to adjust for multiple comparisons we selected proteins which met significance threshold of log p>5 for analyses. With these parameters, I-ASD proteome had nine changes while 16pDel had 48 changes compared to Sib. There were only three proteins (TAPT-1, GGA-1, S100A11) in common between I-ASD and 16pDel proteomes (*Figure 3A*).

In striking contrast, many more changes were observed in the p-proteome analysis: I-ASD NPCs had 105 differentially phosphorylated proteins compared to Sib with 149 unique phosphorylation sites. Further, 16pDel NPCs had 419 differentially phosphorylated proteins with 916 unique phosphorylation sites. Surprisingly, given the absence of genetic overlap between the cohorts, there is considerable overlap between I-ASD and 16pDel p-proteome results with 67 p-proteins (67/105) and 107 phosphorylation sites (107/149) shared between I-ASD and 16pDel (*Figure 3B*). The large differences in the p-proteome along with minimal changes in the proteome suggest that phosphorylation and signaling could be dysregulated in both forms of ASD. Moreover, the p-proteome overlap between I-ASD and 16pDel NPCs suggests that different genetic mutations could converge on common signaling pathways.

To further assess p-proteome data, bioinformatic analyses were performed. To begin to understand the biological processes that could be dysregulated, G-profiler analysis was conducted (*Raudvere et al., 2019*; *Reimand et al., 2019*). For both I-ASD and 16pDel p-proteome, alterations in cytoskeleton, cell cycle, and developmental processes were identified, correlating well with our defined cellular phenotypes. To identify dysregulated signaling pathways, QIAGEN Ingenuity Pathway Analysis

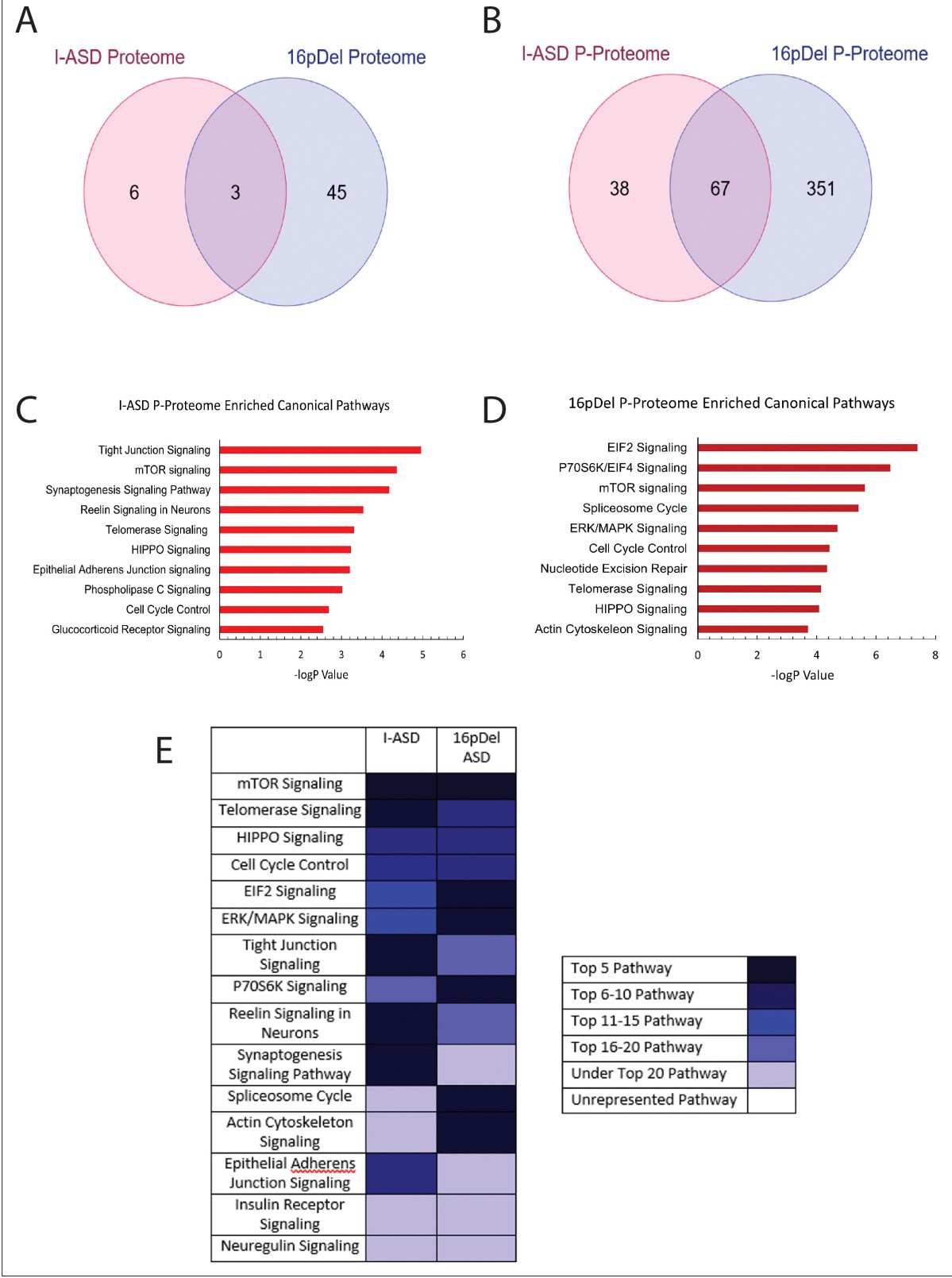

**Figure 3.** Phospho-proteome analysis of I-ASD and 16pDel data. (**A and B**) Venn diagram illustrates proteomic (**A**) and p-proteomic (**B**) changes in both I-ASD and 16pDel NPCs and their overlap. (**C and D**) Ingenuity Pathway Analysis (IPA) canonical pathway analysis of I-ASD (**C**) and 16pDel (**D**) p-proteome data identifies the mTOR pathway as likely disrupted. (**E**) Heatmap of top overlapping canonical pathways between I-ASD and 16pDel p-proteome showing that the strongest overlap between I-ASD and 16pDel p-proteomes is the mTOR pathway. All p-proteome and proteome analyses

*Figure 3 continued*

were done with pooled proteins from all 3 I-ASD, all 3 Sib, and the 2 male 16pDel. Protein was extracted from at least 2 clones and 2 neural inductions per individual.

The online version of this article includes the following source data and figure supplement(s) for figure 3:

**Source data 1.** IPA p-proteome canonical pathway proteins.

**Figure supplement 1.** I-ASD p-proteome network analysis.

**Figure supplement 2.** I6pDel p-proteome network analysis.

**Figure supplement 3.** Ingenuity Pathway Analysis (IPA) total proteome canonical pathway analysis.

**Figure supplement 4.** Ingenuity Pathway Analysis (IPA) of I-ASD whole genome sequencing (WGS).

(IPA) was utilized (*Krämer et al., 2014*; *Schubert et al., 2015*). IPA of I-ASD p-proteome determined the top three enriched modules are tight junction signaling, mTOR signaling, and synaptogenesis signaling (*Figure 3C*, ). For the 16pDel p-proteome, the top three pathways are EIF2 signaling, regulation of EIF4 and p70S6K signaling, and mTOR signaling (*Figure 3D*, ). Comparison analysis indicated the three most overlapping pathways between I-ASD and 16pDel as mTOR signaling, telomerase signaling, and HIPPO signaling (*Figure 3E*).

In terms of canonical signaling pathways, mTOR has the highest enrichment in our datasets. The mTOR pathway includes a cascade of activation of molecules including phosphorylation of AKT and activation of p70S6K and ultimately phosphorylation of S6. Interestingly, IPA expression network analysis of p-proteome reveals strongly altered p-S6 as a central node of convergence in both I-ASD and 16pDel networks along with other mTOR members (*Figure 3—figure supplements 1 and 2*). Thus, despite the two datasets being genetically distinct, both I-ASD and 16pDel share many p-proteome changes with mTOR signaling and convergence onto RPS6 being common between them.

In striking contrast, IPA of total proteome data even with threshold log p > 3 did not find any mTOR enrichment in either I-ASD or 16pDel NPCs (*Figure 3—figure supplement 3*). For I-ASD, we also utilized available WGS data to run IPA. Much like the total proteome results, I-ASD WGS did not show any mTOR pathway enrichment with canonical pathway analyses or with network analysis (*Figure 3—figure supplement 4*). Thus, the strong converging mTOR signaling pathway abnormalities are only uncovered with analysis of the p-proteome.

## mTOR abnormalities are found in all six ASD NPCs

To further assess the mTOR pathway, western blot studies were conducted to define p-S6 levels in 48 hr NPC cultures. Remarkably, p-S6 dysregulation was found in all six ASD NPCs from both cohorts (*Figure 4*). Further, less significant changes were also noted in p-AKT levels in all ASD NPCs (*Figure 4—figure supplement 1*). Importantly, total S6 and AKT protein levels were not different between any of the NPCs. While p-proteome analysis revealed p-S6 as a common dysregulated node among all ASD individuals (*Figure 3—figure supplements 1 and 2*), our western studies revealed two distinct groups of p-S6 dysregulation. In one group, p-S6 levels were reduced relative to controls whereas in the other p-S6 levels were elevated, suggesting two distinct 'signaling subtypes' as described below.

In the I-ASD cohort, Family-1 (I-ASD-1) had a 75% reduction in p-S6 compared to Sib-1 (*Figure 4A and B*) while I-ASD-3 from Family-3 had a 45% reduction in p-S6 compared to Sib-3 (*Figure 4C and D*, p < 0.001). Surprisingly, in contrast, I-ASD-2 from Family-2 had elevated levels of p-S6 (*Figure 4E and F*, 260% increase, p < 0.001) when compared to Sib-2. The abnormal p-S6 levels of each ASD NPC were also significant when compared to the mean of the three Sibs (*Figure 4—figure supplement 2*). Likewise, p-AKT data followed the same pattern but with a smaller effect (*Figure 4—figure supplement 1*). In the 16pDel group, all three 16pDel patient NPCs had higher p-S6 levels compared to both Sib and NIH NPCs (M1: ~170%, p < 0.0001, M2: ~130%, p < 0.001, F: ~80%, p < 0.01 increase, *Figure 4G and H*). There were no statistical differences in p-S6 levels between Sib and NIH NPCs (p = 0.99). All 16pDel NPCs also displayed higher p-AKT levels compared to NIH but not Sib NPCs (*Figure 4—figure supplement 1G and H*).

Thus, western studies validate the p-proteome results in our six ASD patients and confirm changes in both p-S6 and to some extent p-AKT, which can be divided into low mTOR and high mTOR groups. Ultimately, the most striking finding is the consistent mTOR signaling differences across two genetically distinct groups of autism, a finding not previously described.

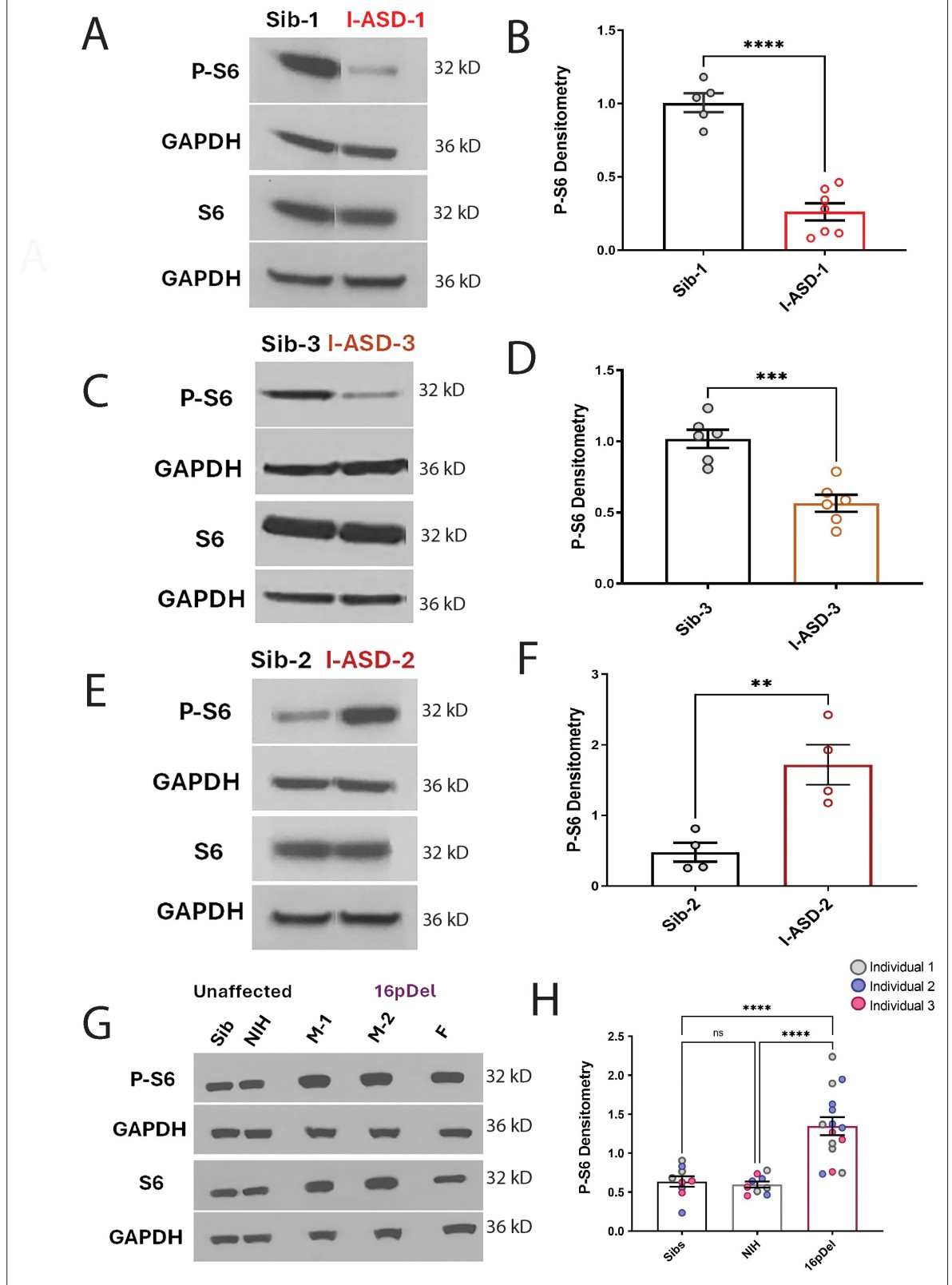

**Figure 4.** P-S6 and S6 western analysis for I-ASD and 16pDel neural precursor cells (NPCs). All images show P-S6 and S6 blots with matched GAPDH loading control. Graphs show densitometry quantifications of normalized p-S6 (p-S6/GAPDH) divided by normalized S6 (S6/GAPDH). Student's t-test utilized for all I-ASD vs Sib comparisons. (**A–D**) I-ASD-1 and -3 representative western blots showing reduced p-S6 but similar S6 and GAPDH in I-ASD-1 compared to Sib-1 (**A**) and in I-ASD-3 vs. Sib-3 (**C**). Graphs (**B and D**): reduced p-S6/S6 in I-ASD-1 vs. Sib-1 (**B**) and in I-ASD-3 vs Sib-3 (**D**). (**E**) I-ASD-2

*Figure 4 continued on next page*

*Figure 4 continued*

representative western blot showing increased p-S6 but similar S6 in I-ASD-2 compared to Sib-2. (**F**) Graph: elevated p-S6/S6 in I-ASD-2 compared to Sib-2. (**G**) Western blot comparing a representative Sib, NIH control, and each 16pDel patient (M-1, M-2, F) showing increased p-S6 in all 16pDel NPCs compared to both NIH and Sib with similar total S6 and GAPDH. (**H**) Graph showing increased p-S6 in 16pDel compared to both Sibs and NIH (one-way ANOVA). For all graphs: p≤0.05 = *, p≤0.01 = **, p≤0.001 = ***, p≤0.0001 = ****. For detailed N values please see *Supplementary file 1*.

The online version of this article includes the following figure supplement(s) for figure 4:

**Figure supplement 1.** p-AKT in I-ASD and 16pdel neural precursor cells (NPCs).

**Figure supplement 2.** p-S6 levels compared across families.

## mTOR pathway modulation rescues and recapitulates common neurodevelopmental phenotypes in low mTOR NPCs

To determine whether the mTOR differences described above drive the developmental abnormalities in our cohorts, we conducted gain and loss of function studies on all I-ASD and both male 16pDel NPCs. As there were no small molecule activator and inhibitor pairs for p-S6 when experiments were being conducted, we instead employed an AKT activator (SC-79) and AKT inhibitor (MK-2206), which have been used extensively in other systems to modulate mTOR pathway (*Jo et al., 2012*; *Takizawa et al., 2012*; *Hirai et al., 2010*). Notably, in the mTOR pathway AKT phosphorylation leads to mTOR pathway activation and ultimately S6 phosphorylation. Given p-AKT was not a central node in our p-proteome analyses and the p-AKT western data readout was less robust, our experiments largely focused on effects of small molecule activator and inhibitor on p-S6. However, we did conduct a brief proof of principle experiment which found that Sc-79 and MK-2206 did increase and decrease (respectively) AKT levels in human NPCS (data not shown).

We first examined our low mTOR cohort comprised of I-ASD-1 and I-ASD-3. Focusing first on Family-1, addition of the SC-79 activator (2 µg/mL) to I-ASD-1 NPCs increased p-S6 levels to that of Sib (*Figure 5A and B*). Importantly, SC-79 at this dose had no effect on p-S6 levels in Sib, nor on the levels of total S6 in either Sib or I-ASD. We next tested whether rescuing p-S6 with SC-79 affected neurite outgrowth or migration. Indeed, SC-79 increased the percentage of neurites (*Figure 5C*) and migration in I-ASD NPCs (*Figure 5D*) without changing these parameters in Sib. Thus, we could rescue the neurite and migration defects in I-ASD-1 by mTOR modulation and increasing p-S6 levels.

If mTOR pathway deficiency is truly contributing to ASD defects in neurite outgrowth and migration, then inhibitor MK-2206 should diminish neurite outgrowth and impair migration in Sib-1 NPCs. MK-2206 (30 nM) reduced p-S6 levels in Sib down to that of I-ASD-1 (*Figure 5E and F*). MK-2206 exposure also reduced the percentage of neurites (*Figure 5G*) and migration (*Figure 5H*) of Sib-1 NPCs to that of I-ASD-1 NPCs thereby reproducing both the ASD neurite and migration defects.

Lastly, we tested whether modulating p-S6 in I-ASD-3, the other 'low mTOR' individual, also led to similar results. Like in I-ASD-1, SC-79 increased p-S6 levels to that of Sib-3 in I-ASD-3 (*Figure 5I*). Likewise, SC-79 also led to an increase in neurite outgrowth in I-ASD-3 that paralleled Sib-3 NPCs (*Figure 5J*). Likewise, inhibiting p-S6 with MK-2206 in Sib-3 (*Figure 5K*) led to a reduction in neurite outgrowth that mimics the phenotype seen in I-ASD-3 NPCs (*Figure 5L*). Thus, in both 'low-mTOR' ASD patients, we find that increasing mTOR pathway activity (via SC-79) can rescue ASD defects while reducing mTOR pathway activity (via MK-2206) in the Sibs reproduces autism NPC phenotypes.

## mTOR pathway modulation rescues and recapitulates common neurodevelopmental phenotypes in high mTOR NPCs

Given that increasing p-S6 in our low mTOR cohort led to rescue of ASD-associated phenotypes, we next sought to determine whether decreasing p-S6 in high mTOR ASD NPCs (ASD-2 and 16pDel) could similarly rescue ASD NPC phenotypes. Starting with I-ASD-2, addition of MK-2206 inhibitor reduced the levels of p-S6 to that of Sib without changing total S6 levels (*Figure 6A and B*). This increased both neurite outgrowth and migration paralleling levels seen in Sib, thereby rescuing the phenotypes (*Figure 6C and D*). Significantly, Sib-2 had no significant response to MK-2206 at this dose.

We next determined if elevating mTOR signaling using SC-79 could reproduce the I-ASD-2 defects in the Sib. SC-79 increased p-S6 levels in Sib-2 NPCs to that of I-ASD-2 (*Figure 6E and F*), which

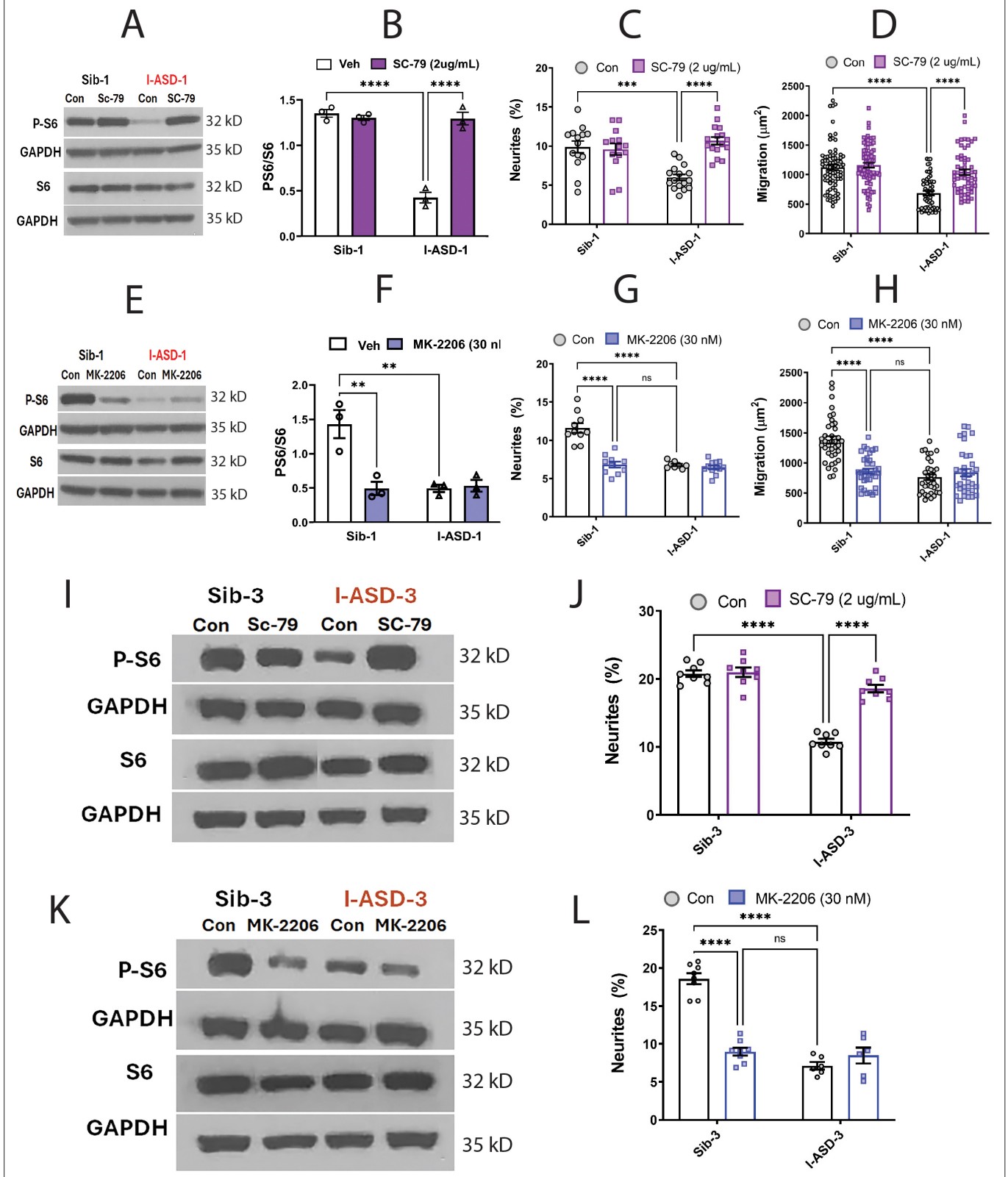

**Figure 5.** Effects of mTOR pathway manipulation in low mTOR cohort. (**A**) Representative western blot: treatment of Family-1 neural precursor cells (NPCs) with SC-79 increases p-S6 levels in I-ASD-1 but not Sib-1 with no changes in total S6. (**B**) Graph: SC-79 vs Veh: increases p-S6/S6 levels in I-ASD-1 but not in Sib-1. (**C**) SC-79 treatment rescues neurite outgrowth in low mTOR I-ASD-1 NPCs without affecting Sib-1 NPC neurites. (**D**) SC-79 treatment rescues migration in low mTOR I-ASD-1 NPCs without affecting Sib-1 NPC neurites. (**E**) Representative western blot: MK-2206 treatment of Family-1

*Figure 5 continued on next page*

*Figure 5 continued*

NPCs decreases p-S6 in Sib-1 but not I-ASD-1 with no changes in total S6. (**F**) Graph: MK-2206 vs Veh: decreases PS6 in Sib-1 but not I-ASD-1. (**G**) MK-2206 treatment reduces neurite outgrowth in Sib-1 to the level of I-ASD-1 without affecting I-ASD-1 neurite outgrowth. (**H**) MK-2206 treatment reduces migration in Sib-1 to the level of I-ASD-1 without affecting I-ASD-1. (**I**) Proof of principle western blot showing that SC-79 increases p-S6 levels in I-ASD-3 but not in Sib-3. (**J**) SC-79 treatment of low mTOR I-ASD-3 increases neurite outgrowth to the level of Sib-3. (**K**) Proof of principle western blot showing that MK-2206 decreases p-S6 in Sib-3 without affecting I-ASD-3 p-S6 levels. (**L**) Treatment of Sib-3 NPCs with MK-2206 reduces neurite outgrowth to the level of I-ASD-3. For all graphs: p≤0.05 = *, p≤ 0.01 = **, p≤0.001 = ***, p≤0.0001 = ****, two-way ANOVA. For detailed N values please see **Supplementary file 1**.

remarkably led to neurite and migration impairments that mimic those seen in I-ASD-2 (*Figure 6G and H*).

Finally, we conducted gain and loss of function studies in the two male 16pDel NPC and NIH NPCs. Like I-ASD-2, the 16pDel NPCs have reduced neurites and migration in conjunction with elevated p-S6. Reducing p-S6 levels with inhibitor MK-2206 (*Figure 6I*) rescued the 16pDel phenotypes as shown by increased neurite outgrowth (*Figure 6J*). Conversely, increasing p-S6 with activator SC-79 (*Figure 6K*) in NIH NPCs reproduced the neurite defects noted in the 16pDel NPCs (*Figure 6L*). In summary, we find that both over- and under-activation of mTOR signaling leads to common reduction in neurites and migration. Manipulation of mTOR could rescue or reproduce ASD phenotypes in both low and high mTOR NPCs, indicating that tight regulation of mTOR pathway is essential for normal neural development.

## Altering the mTOR signaling milieu establishes EF responsiveness in I-ASD NPCs

The above results indicate that mTOR signaling dysregulation is responsible for the shared neurodevelopmental phenotypes. However, one phenotype that diverges between I-ASD NPCs and 16pDel NPCs is responsiveness to EFs. Unlike Sib, NIH, and 16pDel NPCs, the I-ASD NPCs do not respond to EFs (PACAP, NGF, and 5-HT) which stimulate neurite outgrowth and migration in all other NPC subgroups. Given that PACAP, NGF, and 5-HT act through different receptor systems we postulated that dysregulated signaling could potentially contribute to lack of I-ASD response to EFs. Thus, to further investigate if this failure to respond to EFs is due to mTOR signaling defects, we treated the I-ASD NPCs with 'subthreshold' amounts of mTOR agonist and inhibitor which did not maximally increase p-S6 or elicit neurite outgrowth on its own. To do so, we performed parallel dose-response studies and selected drug doses that did not elicit neurite outgrowth on their own but did elicit a non-maximal p-S6 increase (data not shown). For Family-1, agonist SC-79 (0.1 μg/mL) did not stimulate neurite outgrowth in I-ASD-1 NPCs (*Figure 7A*). However, when combined with PACAP, NGF, and 5-HT, the I-ASD-1 NPCs now responded to all three EFs (*Figure 7A*). This rescue effect was specific to I-ASD-1 NPCs as addition of subthreshold SC-79 to Sib NPCs did not heighten EF response (*Figure 7B*). Thus, by using subthreshold doses of mTOR activator, we could facilitate EFs signaling in I-ASD-1 NPCs.

Conversely, to test whether diminishing p-S6 in Sib abolishes EF responses, we took a similar 'subthreshold' approach. Addition of 'subthreshold' MK-2206 (1 nM) to Sib-1 NPCs abolished their typical EF responses (*Figure 7C*).

Finally, we also examined high mTOR Family-2: using a 'subthreshold' dose of inhibitor (1 nM MK-2206), I-ASD-2 NPCs were now able to respond to EFs (*Figure 7D*). These 'subthreshold' studies further support altered mTOR signaling in NPCs and that manipulation of mTOR signaling can alter ASD and control NPC response to EFs during neurodevelopment.

## Discussion

In this study, we utilized NPCs derived from 12 different individuals: six ASD cases, from two different ASD subtypes (idiopathic and 16p11.2 deletion) and two different control groups (Sib and NIH). Unexpectedly, we found common deficits in neurite outgrowth and cell migration in all six ASD NPCs. Yet, WGS revealed that I-ASD and 16pDel individuals did not share any rare variants in the 28 genes deleted in the 16p11.2 CNV and the three I-ASD individuals share no rare functional protein-coding variants that were ASD specific. Despite the genetic heterogeneity within the I-ASD cohort and between

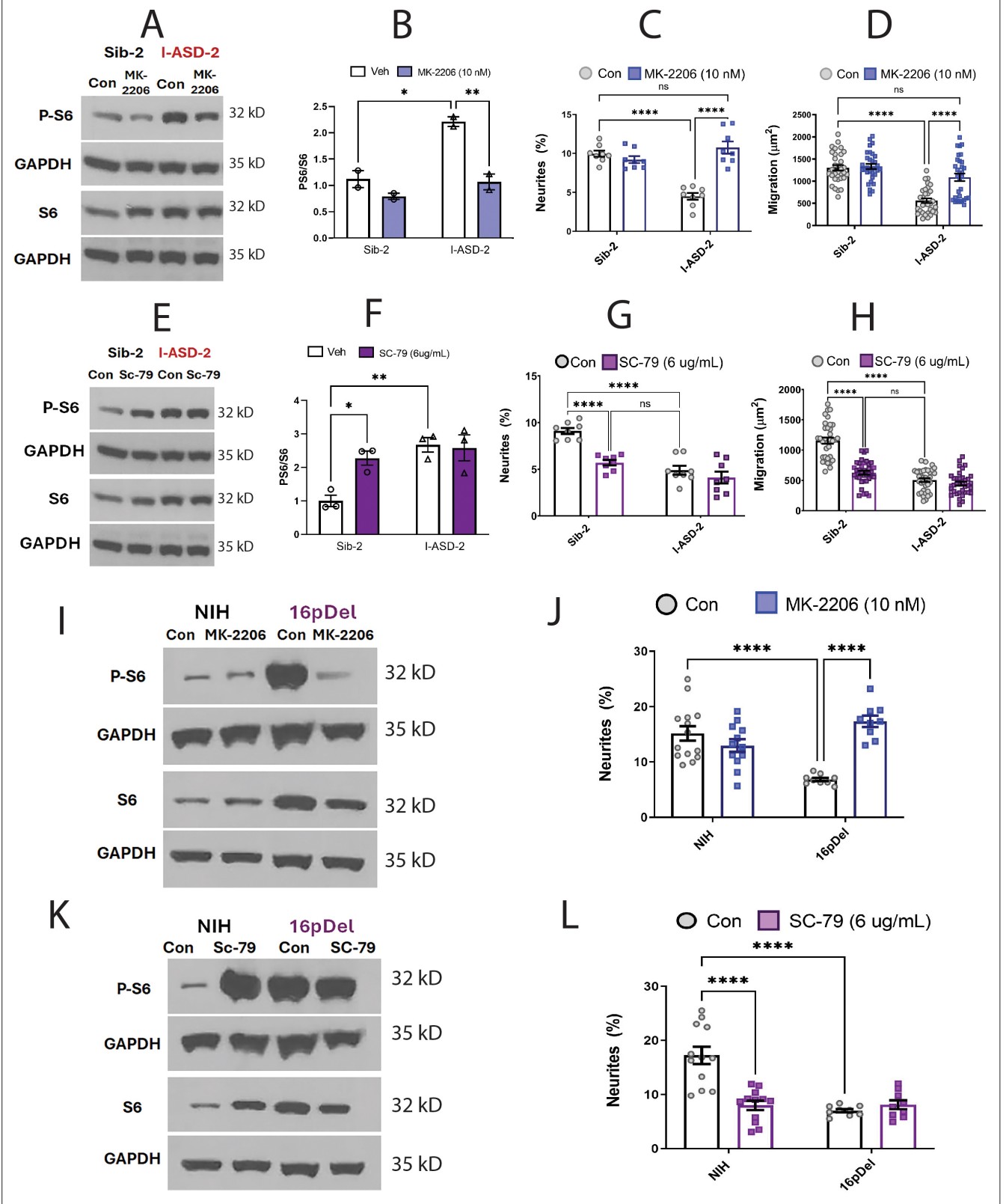

**Figure 6.** Effects of mTOR pathway manipulation on high mTOR cohort. (**A**) Representative western: MK-2206 treatment of I-ASD-2 neural precursor cells (NPCs) leads to reduction of p-S6 in I-ASD-2 to Sib-2 levels without changing total S6. (**B**) Quantification of PS6/S6 western blots showing decreased PS6 in I-ASD-2 but no change in Sib-2 with MK-2206 treatment. (**C**) MK-2206 rescues neurite outgrowth in high mTOR I-ASD-2 without changing Sib-2 neurites. (**D**) MK-2206 rescues migration in high mTOR I-ASD-2 without changing Sib-2 migration. (**E**) Representative western: SC-

*Figure 6 continued on next page*

*Figure 6 continued*

79 treatment of Family-2 NPCs increases p-S6 in Sib-2 to the level of I-ASD-2 without changing total S6. (**F**) Quantification of Veh vs.SC-79 westerns showing increased PS6 in Sib-2 but no change in I-ASD-2. (**G**) Treatment of Sib-2 with SC-79 diminished neurite to the level of I-ASD-2. (**H**) Treatment of Sib-2 with SC-79 diminished migration to the level of I-ASD-2. (**I**) Proof of principal western: MK-2206 treatment of 16pDel NPCs leads to reduction of p-S6 in 16pDel NPCs to NIH levels without changing total S6. (**J**) In high mTOR 16pDel NPCs, MK-2206 treatment rescues neurites without significantly affecting NIH NPCs. (**K**) Proof of principle western: SC-79 treatment of NIH NPCs leads to increase in p-S6 to the level of 16pDel NPCs without changing total S6. (**L**) Treatment of NIH NPCs with SC-79 diminishes neurite outgrowth to 16pDel levels without affecting 16pDel NPCs. For all graphs: p≤0.05 = *, p≤0.01 = **, p≤0.001 = ***, p≤0.0001 = ****, two-way ANOVA. For detailed N values please see ***Supplementary file 1***.

I-ASD and 16pDel cohort, p-proteomic analyses show significant overlap between I-ASD and 16pDel with convergence onto the mTOR pathway. Subsequent western analyses validated these p-proteome differences and uncovered two distinct mTOR molecular subtypes, characterized by increased or decreased p-S6. Importantly, the mTOR differences are responsible for the shared neurodevelopmental phenotypes since manipulation of the mTOR pathway with small molecules could rescue the autism NPC deficits and reproduce autism NPC phenotypes in control NPCs. Lastly, this is one of the first studies to utilize EFs to uncover differences not apparent in control conditions which subsequently allowed for definition of two further subtypes of ASD in our dataset, EF responsive (16pDel) and EF unresponsive (I-ASD). Remarkably, mTOR manipulation also rescued EF non-responsiveness indicating the importance of this pathway as a driver of disease phenotypes. Thus, our study supports that ASD subtypes with non-overlapping genetics can have common dysregulation of mTOR signaling which is responsible for the converging common neurodevelopmental phenotypes.

## Altered neurite outgrowth and migration are a common phenotype in all six ASD NPCs

While most ASD iPSC studies have investigated terminal neuronal differentiation phenotypes, our study has focused on NPCs and may be the first to find common NPC neurite and migration impairments in two different ASD subtypes. Previous iPSC-based studies of NDDs, such as Fragile X syndrome, 22q13 deletion syndrome, 16p11.2 deletion and duplication syndromes, and macrocephalic I-ASD patients have noted alterations in dendrites, axons, and neurites in post-mitotic neurons (***Bhattacharyya and Zhao, 2016***; ***Castrén et al., 2005***; ***Doers et al., 2014***; ***Shcheglovitov et al., 2013***; ***Kathuria et al., 2018***; ***Deshpande et al., 2017***; ***Griesi-Oliveira et al., 2015***; ***Marchetto et al., 2017***). Migration defects have not been commonly noted in ASD but have been reported in studies of NDDs including schizophrenia, Rett syndrome, Down syndrome, Timothy syndrome, and CNTNAP2 mutation (***Brennand et al., 2015***; ***Flaherty et al., 2017***; ***Krey et al., 2013***; ***Huo et al., 2018***; ***Zhang et al., 2016***; ***Birey et al., 2022***). Thus, our studies provide further evidence to altered developmental neuronal phenotypes in ASD.

Multiple studies indicate that NPCs are a biologically relevant cell type for ASD. Genetic studies, including extensive WES and GWAS analysis, have reproducibly identified human mid-fetal cerebral/cortical (8–24 weeks) development as a window for when and where most ASD risk genes are expressed (***Satterstrom et al., 2020***; ***Willsey et al., 2013***; ***Parikshak et al., 2013***). During this window, NPCs are formed and undergo migration and early neurite extension. Recently altered NPC development has also been identified as a consistent neurodevelopmental phenotype for numerous ASD risk genes (***Garcia-Forn et al., 2020***). Given that human iPSC-derived NPCs are similar transcriptionally and functionally to cortical NPCs, our studies provide experimental evidence of NPC developmental dysregulation in ASD (***Hofrichter et al., 2017***; ***Yan et al., 2013a***). Furthermore, by comparing two subtypes of ASD, we show, as suggested by genetics studies, that this early developmental dysregulation may be a common feature among multiple forms of ASD.

## mTOR signaling as a point of convergence in ASD

Our p-proteome, westerns, and small molecule analyses identified altered mTOR signaling as a point of convergence for the shared ASD neurite and migration phenotypes. To our knowledge, we are the first to utilize unbiased p-proteomics in ASD NPCs and were able to uncover marked differences in the p-proteome for both ASD subtypes when compared to control. Interestingly, despite non-overlapping genetics, significant overlap was observed (67/105) at the p-proteome level between I-ASD and 16pDel. Further, about a third of these p-proteins are common with 102 high confidence

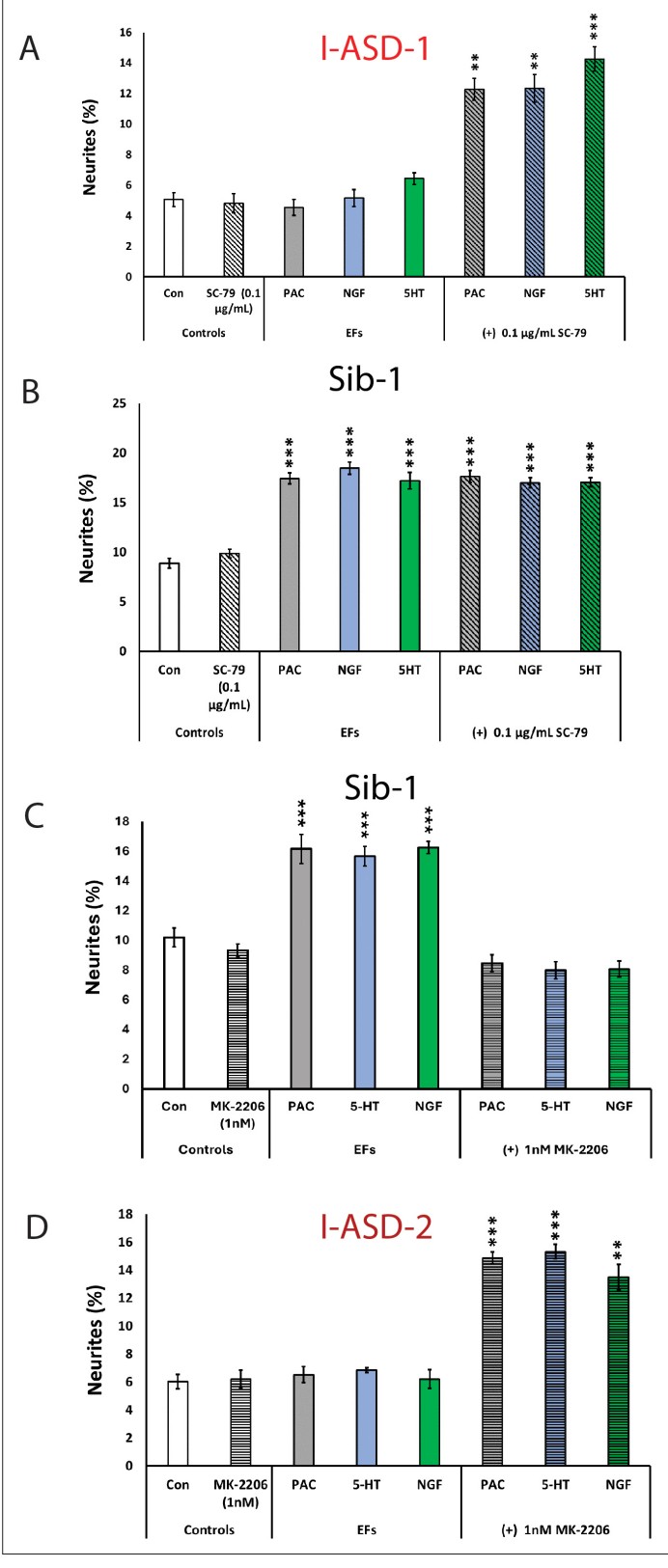

**Figure 7.** Modulation of mTOR can facilitate or abolish extracellular factor (EF) responses in I-ASD. (**A**) Treatment of low mTOR I-ASD-1 neural precursor cells (NPCs) with subthreshold dose of SC-79 (0.1 µg/mL) results in NPCs responding to EFs. (**B**) Treatment of Sib-1 NPCs with a subthreshold dose of SC-79 does not alter EF response. (**C**) Treatment of Sib-1 NPCs with subthreshold MK-2206 (1 nM) abolishes NPC response to EFs. (**D**) Treatment of

*Figure 7 continued on next page*

*Figure 7 continued*

high mTOR I-ASD-2 with subthreshold MK-2206 (1 nM) establishes EF responses. p < 0.001 for all comparisons, one-way ANOVA. For all graphs: p≤0.05 = *, p≤0.01 = **, p≤0.001 = ***, p≤0.0001 = ****. For detailed N values please see **Supplementary file 1**.

ASD risk genes or relevant gene families (*Satterstrom et al., 2020*), suggesting that we are identifying similar pathways as described by others. In contrast, very few proteomic changes were noted in I-ASD (9/9700) and 16pDel (48/9700) NPCs when compared to unaffected individuals. This could suggest either the proteome is relatively unaltered in ASD NPCs, or the changes are quite varied between samples so no consistent significant results were observed.

The convergent signaling results are even more remarkable when we consider the WGS results and total proteome results. There is no rare variant overlap between the two ASD datasets, or among the three I-ASD individuals. While some I-ASD individuals have rare variants in genes that could affect mTOR signaling, these were often also shared by the Sib (e.g. TSC-1 rare variants). Given that the non-overlapping variants could potentially converge onto the mTOR pathway, we subsequently ran IPA on WGS data which did not uncover convergence onto the mTOR pathway or any other classical signaling pathway such as ERK, WNT, or CREB, but did find common alterations in immune regulatory pathways (*Figure 3—figure supplement 4*). Likewise, pathway analysis of published transcriptome data from 16pDel ASD patient iPSCs or lymphoblastic tissue also reveals no mTOR enrichment or dysregulation in signaling (*Blumenthal et al., 2014*). These results are also mirrored in the total proteome analyses which do not suggest mTOR enrichment (*Figure 3—figure supplement 3*). These results suggest that even without genetic mutations or protein changes in the mTOR pathway, multiple rare and common variant genes could interact to lead to dysregulation of mTOR pathway signaling. Moreover, the presence of high and low mTOR I-ASD NPCs could conceivably be dependent upon the specific type of genetic mutations present and their effect on mTOR signaling. Consistent with this possibility, all three 16pDel individuals who share the same CNV deletion have NPCs with high mTOR signaling. Ultimately, genetic analyses alone did not reveal convergence onto the mTOR pathway or alteration in the signaling milieu in our cohort. We were only able to detect our remarkably convergent mTOR deficits and alterations in signaling by utilizing p-proteomics and studying cellular signaling through western blotting, further indicating that ASD could be a disorder of intracellular signaling.

Consistent with the idea that mTOR signaling dysfunction could be a common mechanism for autism, monogenic syndromic disorders, which can exhibit ASD such as tuberous sclerosis (TSC), PTEN-associated ASD, and neurofibromatosis-1, have mutations that directly affect mTOR pathway components (*Enriquez-Barreto and Morales, 2016*; *Yeung et al., 2017*; *Huber et al., 2015*; *Dasgupta et al., 2005*; *Winden et al., 2019*; *Sharma et al., 2010*; *Rangasamy et al., 2016*; *Costales and Kolevzon, 2015*). Furthermore, some studies of syndromic diseases where the driving mutation is not in the mTOR pathway, such as Fragile-X, Angelman, Rett syndrome, and Phelan-McDermid syndrome (22q13 deletion), also show mTOR alterations in human postmortem samples, embryonic stem cell models, and hiPSC-based models (*Winden et al., 2018*). In 16pDel, because the MAPK3 gene is deleted, both rodent and human studies have uncovered alterations in ERK1 signaling (*Pucilowska et al., 2018*; *Pucilowska et al., 2015*; *Deshpande et al., 2017*). Importantly for 16pDel, altered mTOR has not been reported previously. On the other hand, there are no iPSC studies of I-ASD that show mTOR abnormalities. However, recent studies in peripheral white blood cells identified mTOR signaling proteins and p-proteins as being predictors of not only ASD diagnosis but also disease severity (*Onore et al., 2017*; *Rosina et al., 2019*). Yet, it is unclear whether WBC mTOR alterations are sufficient to suggest that mTOR is contributing to neurodevelopmental phenotypes in these I-ASD individuals. Regardless, our study adds to the growing literature that mTOR signaling dysfunction is a shared pathogenic mechanism among different types of ASDs and raises the possibility that mTOR dysregulation could be observed commonly across multiple additional ASD subtypes. Thus, studying mTOR in multiple ASD subtypes could be important in deciphering the molecular etiology of ASD.

Of course, while mTOR seems to play an important role in multiple subtypes of ASD, numerous other signaling pathways have also been implicated in the pathogenesis of ASD. As noted above, in 16p11.2 deletion, ERK/MAPK3 is part of the 11.2 segment and implicated in driving some of the neurodevelopmental phenotypes associated with the deletion and duplication syndromes. The WNT pathway, which includes CHD8 (an autism-related chromatin modifier) and beta-catenin, has long

been implicated in ASD pathogenesis. The WNT pathway, much like mTOR, is thought to regulate neuronal proliferation and has been associated with macrocephalic phenotypes seen in ASD. For example, a study by Marchetto et al. found that NPCs derived from individuals with macrocephaly and I-ASD displayed excessive proliferation as well as altered WNT signaling. In our p-proteome analyses, the ERK/MAPK and PKA pathway have also been shown to be enriched. Ultimately, altered cellular signaling seems to play an important role in the pathogenesis of ASD and other NDDs. Importantly, our study indicates that ASD NPCs not only have enrichment of dysregulated mTOR but that mTOR dysregulation is driving autism-associated phenotypes.

## The mTOR pathway and EF responses

The importance of mTOR signaling was further highlighted by our small molecule gain and loss of function analyses and EF studies. By correcting the p-S6 levels with mTOR agonists and antagonists, we were able to rescue the developmental defects in both I-ASD and 16pDel NPCs. Furthermore, by changing Sib control p-S6 to ASD levels, the neurodevelopmental defects could be reproduced demonstrating that the mTOR signaling differences are mechanistically linked to the shared neuro-developmental defects for both I-ASD and 16pDel NPCs. In addition, as one of the first studies to challenge ASD NPCs with EFs, we were able to uncover the importance of mTOR in the regulation of the signaling milieu and uncover differences between I-ASD and 16pDel in their ability to respond to EFs. Specifically, by utilizing subthreshold doses of SC-79 and MK-2206, we were able to facili-tate responses to EFs in all I-ASD NPCs. These results suggest that mTOR homeostasis and ambient levels may be critical for facilitating several cellular processes including the ability for other signaling pathways to function effectively. As 16pDel NPCs had no EF response impairments, different ASD subtypes or different genetic backgrounds may have distinct thresholds for maintaining signaling homeostasis. Our studies point strongly to the importance of the signaling milieu and mTOR regula-tion in the control of neurodevelopment and ASD pathogenesis.

Interestingly, in I-ASD NPCs, mTOR seems to play a pivotal role in EF responsiveness. Both 'high' and 'low' I-ASD NPCs fail to respond to EFs while altering mTOR milieu with subthreshold doses of MK-2206 or SC-79 allows the I-ASD NPCs to respond to EFs. Yet, remarkably, 16pDel NPCs, which share elevated mTOR signaling with I-ASD-2 NPCs, have no deficits in EF responsiveness. This suggests that mTOR dysregulation alone is not sufficient to cause EF deficits in all NPCs. While there are several commonalities between 16pDel NPCs and I-ASD NPCs, our proteomic analyses show that there are significant differences as well. Specifically, the I-ASD p-proteome has 38 differently phosphorylated proteins that are not shared with the 16pDel and the 16pDel p-proteome has 351 differently phosphorylated proteins not shared with the I-ASD NPCs. Analysis of these 'differently' expressed p-proteins show that I-ASD NPCs seem to have dysregulation in tight junction signaling as well as in proteins such as spectrin and ankyrin which are essential to appropriate stabilization of membrane proteins (such as signaling receptors) to the underlying cytoskeleton. Thus, it is possible that interaction of mTOR defects with tight junction/anchoring molecules lead to the lack of responses to numerous EFs (all of which work on different receptor systems). We have not conducted specific studies into the difference between I-ASD and 16pDel when it comes to EF responsiveness, though the p-proteome results provide the above tantalizing clues which could be focused on in future research. Regardless, the most interesting point is that despite many commonalities, I-ASD and 16pDel NPCs do not share the EF unresponsiveness phenotype, though for I-ASD NPCs, EF responsiveness can be altered by targeting mTOR signaling.

## EF responses and ASD

In the developing brain, EFs regulate basic developmental processes like neurite outgrowth and migration and alterations in EFs have been implicated in the pathogenesis of NDDs (*Otto et al., 2001*; *Hashimoto et al., 2001*; *Cameron et al., 2007*; *Lu and DiCicco-Bloom, 1997*; *Yan et al., 2013b*: *DiCicco-Bloom, 1996*; *Nicot and DiCicco-Bloom, 2001*; *Korsching et al., 1986*; *Rajakumar et al., 2004*; *Lazar et al., 2008*; *Bonnin et al., 2007*; *Temple and Qian, 1995*; *Speranza et al., 2015*; *Boukhris et al., 2016*; *Veenstra-VanderWeele et al., 2012*; *Yang et al., 2014*). In our study, we selected three EFs, PACAP, NGF, and 5-HT, known to be expressed during the 'critical window' of mid-fetal development, with well-established roles in regulation of early neurodevelopmental phenotypes to stimulate I-ASD and 16pDel NPCs in both our neurite outgrowth and neurosphere

assays to assess for ASD-specific response differences. Strikingly, NPCs derived from all three idiopathic patients are unresponsive to all three EFs tested, while all the unaffected patient NPCs had increased neurite outgrowth or migration in response to EFs. Such differential response to EFs has been noted in previous work in our lab on the Engrailed-2 KO model of developmental disorders (*Rossman et al., 2014*). Though studies are limited, further exploration may show that differential responses to regulatory molecules may be a characteristic of some types of developmental disorders. Indeed, meta-analyses have found that about 30% of children with ASD have elevations in blood 5-HT. This increased 5-HT could be due to dysregulated 5-HT production, enhanced uptake, impaired clearance, or a compensatory response to decreased sensitivity to 5-HT in the ASD brain. At the same time, our studies found that unlike I-ASD NPCs, 16pDel NPCs have a typical response to all three EFs with increases in neurite outgrowth and migration. Response to EFs could also have implications for the treatment of individuals with ASD, for example, SSRI are commonly used to treat comorbid depression/anxiety in ASD, but if these individuals have neural cells that are less responsive to 5-HT, this treatment could be less effective, though further research is needed to understand whether NPC responses to EFs/drugs correlate with drug responses in the individual. In sum, while different subtypes of ASD can have common underlying neurobiology, there are subtypes or perhaps even patient-specific differences in the defects seen in ASD. By conducting experiments with EFs we were able to find two distinct subgroups of ASD which would not have been apparent if we had conducted our experiments in control conditions alone, showing the utility of studying patient-derived cells under different conditions.

## Subtyping ASD

The heterogeneity of ASD, the subjective nature of clinical categorization, and the current inability to stratify and subtype the disorder have been postulated as reasons for the decades of unsuccessful clinical trials in ASD. Effectively subtyping ASD would help us uncover the multiple pathways through which ASD can manifest in different individuals, allow us to take a more 'personalized' approach to etiological studies, and could help us understand which therapeutics would be best suited to certain ASD subpopulations. As our study is one of the first to compare two disparate forms of ASD, it has given us insight into both convergent and personalized phenotypes in ASD populations.

Despite common neurodevelopmental phenotypes, both a 'low' and 'high' mTOR signaling subtype is present in our ASD NPCs. This phenomenon of 'too much' or 'too little' of a pathway or gene leading to similar outcomes has been seen before in MECP-2 gain and loss of function as well as in Timothy syndrome (*Birey et al., 2022*; *Peters et al., 2013*; *Ramocki et al., 2010*). Our gain and loss of function studies showed that mTOR dysfunction drives impaired neurodevelopment. Thus, characterizing ASD by mTOR levels could help us identify which groups may be candidates for mTOR activators vs inhibitors in future drug trials. Another subtype-specific difference was that I-ASD individuals failed to respond to EFs while 16pDel ASD individuals had typical increases in neurites and migration under EF stimulation. This difference in EF response did not correlate with the mTOR defects. Specifically, both I-ASD-2 and 16pDel had 'high' mTOR signaling but only I-ASD-2 failed to respond to EFs, indicating that our EF studies help us glean more information than molecular studies alone. While there are no medications that target the core symptoms of ASD, we do use drugs that modulate monoamines such as serotonin (5-HT) and dopamine to manage certain behavioral symptoms in ASD. Furthermore, comorbid conditions such as depression are common in ASD and notoriously difficult to treat with our usual medications (*Williams et al., 2010*). Thus, understanding whether certain subtypes of ASD are responsive to EFs like 5-HT or even medications by using iPSC-derived cells may help us better tailor treatments for ASD and its comorbid conditions.

## Study limitations and future directions

Our study is one of the first to show the remarkable convergence of neurodevelopmental and molecular phenotypes in two distinct autism subtypes using rigorous methods such as utilization of multiple iPSC clones and neural inductions. The extensive rigor of our analyses that included a total of 29 distinct iPSC clones and 61 distinct neural inductions lends to relatively small sample sizes (though similar to other iPSC studies) with only six autism probands, three of each subtype. Thus, it is difficult to know whether our observed phenotypes are generalizable to all individuals with 16pDel, I-ASD, or other ASDs. However, it is important to note that our sample had significant clinical and genotypic

heterogeneity yet, we could still uncover common phenotypes. Expanding our study design to other subtypes of ASD or other NDDs would help establish if common underlying mechanisms could be identified (*Gandal et al., 2018*).

While our studies reveal important alterations in early neurodevelopment in ASD, our monolayer NPC studies do not recapitulate the 3D nature or complex interactions occurring in the developing brain. Moreover, as hiPSC-derived NPCS are most similar to fetal cortical radial glial cells, the phenotypes observed in our dishes may not directly parallel the structural or functional defects in the ASD brain. Neuropathological studies of ASD have indicated brain defects suggestive of dysregulated development. However, studies have not been conducted to determine whether, for example, a defect in migration in ASD iPSC-derived NPCs is correlated with altered cortical lamination in the patient from whom the cells were derived. To understand this, MRI and postmortem analyses need to be conducted to correlate iPSC results with neuropathological alterations. Interestingly, an iPSC study of ASD patients with MRI-defined macrocephaly found that affected NPCs proliferated faster than control NPCs (*Marchetto et al., 2017*; *Connacher et al., 2022b*), suggesting correlation between structural MRI and NPC phenotypes.

Our data potentially suggests that mTOR could serve as a target to employ precision medicine techniques in ASD (*Sato, 2016*). However, given that NPCs that are fetal in nature, it is unclear whether targeting mTOR in a child or adult would treat disease phenotypes. Importantly, however, we know that signaling pathways such as mTOR continue to play roles in the postnatal and adult brain (*Pagani et al., 2021*). For example, mTOR is critical for the regulation of processes such as the moment-to-moment modification of dendritic spines which are essential to learning, memory, and integration of information in the brain (*Ganesan et al., 2019*; *Nicolini et al., 2015*; *Tordjman et al., 2015*; *Henry et al., 2017*). Thus, it is reasonable to postulate that mTOR defects, which in the prenatal period leads to altered neurites or migration, could in the adult brain cause altered dendritic functioning and plasticity which could consequently lead to behavioral symptoms of ASD. As such, targeting mTOR in the adult brain could alleviate ASD symptoms. Preclinical mouse models of Rett syndrome support this concept, as treatment of adult animals with mTOR-activating IGF-1 reduced anxiety levels and increased exploratory behaviors, suggesting that targeting mTOR can have an effect outside of the fetal developmental window (*Castro et al., 2014*). On the other hand, clinical trials of IGF-1 in children with Rett syndrome and rapamycin in children with TSC have not thus far had promising effects on behavior (*Khwaja et al., 2014*, *Sahin M, 2018b*, *Sahin M, 2018a*). While early neurodevelopmental processes may no longer be occurring in the postnatal brain, studying NPCs can provide insight into the initial processes that contribute to ASD pathogenesis.

## Materials and methods

### Patient cohorts

#### Idiopathic autism (I-ASD)

I-ASD patients were selected from a larger cohort of 85 New Jersey families recruited by the Brzustowicz laboratory as part of the New Jersey Language and Autism Genetics Study (NJLAGS). In these families, at least one individual meets the criteria for ASD and at least one other family member meets criteria for language-based learning impairment (LLI). Each family member was extensively phenotyped by the same set of clinicians with a battery of behavioral tests as described in *Bartlett et al., 2012*; *Bartlett et al., 2014*. The ASD proband was required to meet the criteria for 'autistic disorder' on two of the three following measures: (1) Autism Diagnostic Interview-Revised (ADI-R), (2) Autism Diagnostic Observation Scale (ADOS), and Diagnostic and Statistical Manual IV (DSM-IV). Moreover, the ASD individual had no known genetic causes of autism such as Fragile X or Rett syndrome. The LLI individual was identified by ruling out ASD, hearing impairments, and other neurological disorders and by using multiple language assessments including the Comprehensive Test of Language Fundamentals (*Bartlett et al., 2012*; *Bartlett et al., 2014*). In addition, each family member underwent 3–5 hr of direct behavioral testing including members of the family who do not have LLI or ASD. All family members were also assessed with the DSM-IV and ADOS to ensure that unaffected Sib and LLI individuals did not meet criteria for autism. From this broader cohort, we selected eight families with the following characteristics: (1) ASD proband with moderate or severe symptoms, (2) families with sex-matched unaffected Sib. The studies presented are on three randomly selected families from this

smaller cohort of eight. Blood samples were obtained from all members of these families and lymphocytes that were cryopreserved were used to generate iPSCs.

For clinical characterization of our I-ASD cohort, please refer to the latest publication from our lab which features clinical data for each individual (*Connacher et al., 2022a*, *Figure 1*). Briefly, I-ASD-1 has severe cognitive impairment (unable to complete IQ test) and ADOS and ADI-R revealed limited language comprehension to a few single words as well as occasional echolalic and scripted speech. SRS score was 90, indicating severe social impairment. I-ASD-1 had 78th percentile head circumference at 4.1 years, indicating normal head size. I-ASD-2 also has severe cognitive impairment (unable to complete IQ test) with ADOS and ADI-R revealing limited language comprehension to single words and directions and almost no language production. The SRS score was 83 (severe social impairment) and proband had a head circumference of 97th percentile at time of measurement (14 years) consistent with macrocephaly. Lastly, I-ASD-3 has limited language comprehension to a small number of single words, Non-verbal IQ (NVIQ) of 118 and SRS score of 69 (moderate social impairment). Unaffected individuals were diagnostically determined to not have any language or learning impairment or ASD as noted above.

## 16p11.2 deletion and NIH controls

Fibroblasts and lymphocytes were acquired from two males and one female patient with the 16p11.2 deletion and autism. These individuals were derived from the Simon's Foundation VIP cohort, now termed Simon Searchlight Collection. Identity codes from SFARI for the individuals are as follows: Female (14,758.x3), Male-1 (14,824.x13), and Male-2 (14,799.x1). ASD inclusion criteria for 16pDel probands required if they meet ADI-R and ADOS score cutoff criterion for ASD or autism (some individuals were clinically assessed using DSM-IV criteria). The probands were assessed on verbal and nonverbal cognitive abilities as described previously (*Simons Vip Consortium, 2012*). As we reported in *Connacher et al., 2022b*, the first male (16pDel-1) exhibited ASD (Asperger's disorder), with a full-scale IQ (FSIQ) of 122, (non-verbal IQ [NVIQ] 130, verbal IQ [VIQ] 106), head circumference at the 99th percentile at 14.5 years, consistent with macrocephaly, and comorbid expressive language disorder, anxiety, and microphthalmia with an SRS score of 76. The second male (16pDel-2) exhibited autism, with FSIQ score of 93 (NVIQ 98, VIQ 87) and a head circumference at the 99th percentile at age 14.3 years, consistent with macrocephaly. He also displayed numerous other developmental phenotypes including coordination disability, developmental delay, cerebral palsy, ADD/ADHD, articulation disorder, and repetitive/expressive language disorder and an SRS score of 90. Lastly 16pDel-F exhibited atypical autism (previously PDD-NOD) with an FSIQ of 71 and NVIQ of 82, with a head circumference of 52.5 at 7.2 years (80th percentile) as well as comorbid articulation disorder, communication disorder, and ADHD.

For 16pDel two iPSC clones per individual were obtained from RUCDR Infinite Biologics (now Sampled). As genetically matched Sibs were unavailable for 16pDel, three sex-matched iPSC control lines from genetically normal newborns from the NIH Regenerative Medicine Program were obtained. Specifically, two male (NCRM-1 and NCRM-3) and one female (NCRM-6) research grade line were utilized. The NIH control iPSC were generated from cord blood using episomal plasmid reprogramming method. https://commonfund.nih.gov/stemcells/lines.16pDel were also compared to the Sibs from I-ASD cohort.

## Derivation, validation, and maintenance of iPSC lines

Sib and I-ASD iPSCs were generated by the Lu & Millonig labs while the 16pdel iPSCs were generated by SAMPLED. For I-ASD and Sib, T lymphocytes were isolated from cryopreserved samples and these cells were expanded and then reprogrammed with a non-integrating Sendai virus containing the four Yamanaka factors: SOX2, OCT3/4, KLF, and a temperature-sensitive C-MYC (*Seki et al., 2011*). The 16pdel iPSCs were also made using the same Sendai virus method, however, both lymphocytes and fibroblasts were used as the initial somatic tissue. iPSCs were characterized using immunocytochemistry (ICC) and QRTPCR for the following markers: NANOG, OCT4, TRA-1–60, SSEA4, CD24, and E-Cadherin. iPSCs were assayed for chromosomal abnormalities via G-band karyotype assay or via Array Comparative Genomic Hybridization (Company: Cell Line Genetics); DNA fingerprinting was also conducted to confirm genetic identity to original tissue samples. See *Connacher et al., 2022a*, Materials and methods, and supplementary information section for details and data on these iPSC characteristics. iPSC lines

were maintained in mTeSR1 media (STEMCELL Technologies, 85850) on six-well plates (Corning, COR-3506) coated with hESC qualified Matrigel (Corning, 354277). mTeSR media was treated with Primocin (antimicrobial) at 1:500 dilution. Media was changed daily. Once cells reached 70–90% confluency, they were passaged by incubating cells with 0.5 mM filter-sterilized EDTA diluted in 1× PBS for 10–20 min. When iPSCs detached from the plate, they were lifted and centrifuged at 150×*g* for 5 min. The cell pellet was then gently resuspended in mTeSR and plated at 15–30% confluence in mTeSR media with 5 µM rock inhibitor (Y-compound), Y27632 (STEMCELL Technologies, 72302) for 24 hr.

## Generation, validation, and maintenance of NPCs

To generate NPCs, iPSCs were induced using a modified version of the Thermo Fisher GIBCO Neural Induction Protocol (*Williams et al., 2018*). In brief, iPSCs at 70–90% confluency were dissociated using 1× Accutase (Thermo Fisher Scientific A111050) and were centrifuged at 150×*g* for 5 min. The resulting pellet was then resuspended into Neural Induction Media (NIM) (Thermo Fisher Scientific, A1647801) with 5 µM of Y-compound. The following densities of iPSCs were plated into Matrigel-coated 12-well plates containing 1 mL of NIM (with 5 µM Y-compound): 80, 120, 200, 300 K. Two wells of each density were made. One set of wells at each density were induced for 7 days while the other was induced for 8 days. Once induction day 7 or 8 was reached, cells were lifted with 1× Accutase and centrifuged at 300×*g* and then resuspended in Neural Expansion Media (NEM) (Thermo Fisher Scientific, A1647801). 1–1.5 million cells were then replated onto Matrigel-coated six-well plate with 2 mL NEM with 5 µM Y-compound. At this point cells are P0. For our studies, NPCs were used only from P3 to P8. At P3-P5 lines were tested with Lonza mycoplasma detection kit to ensure that media/cells were contamination free. All media utilized was supplemented with primocin (antimicrobial) at 1:500 dilution.

To assess NPC identity, ICC was conducted to verify that cells expressed the following NPC markers (Sox2, Pax6, and Nestin). NPCs were discarded if marker immunostaining revealed Nestin or Sox2 <85% or Pax6 <60% (*Connacher et al., 2022a*). NPCs were also stained with OCT3/4 to ensure they no longer expressed this iPSC marker. As shown in *Connacher et al., 2022b*, Quantiplex analysis was also performed on NPC mRNA to assess expression of several NPC markers.

NPCs were also differentiated into neurons, astrocytes, and oligodendrocytes as per Thermo Fisher Protocol: https://www.thermofisher.com/us/en/home/references/protocols/neurobiology/neurobiology-protocols/differentiating-neural-stem-cells-into-neurons-and-glial-cells.html. Differentiated cell identity was assessed by ICC for the following markers: MAP2/Tau (Neuron), O4/Gal-c (oligodendrocytes), and GFAP for astrocytes (*Connacher et al., 2022a*).

## WGS and variant calling

WGS data for the samples in the I-ASD dataset was extracted from a previous study (*Zhou et al., 2023*). The details of sequencing and variant identification are described previously (*Zhou et al., 2023*). The sequencing data and the variant calls are available in the National Institute of Mental Health Data Archive (NDA) under projects C1932 and C2933.

The variants were annotated with ANNOVAR (ver. 2017-07-17, human reference build hg19) (*Wang et al., 2010*) to obtain the mutation effect, population allele frequency (AF), etc. After annotation, variants were selected based on the following criteria: (1) have a 'PASS' value in the FILTER field; (2) have an AF <5% in database gnomAD_genome_ALL; (3) exonic variants with the functional categories of 'non-synonymous', 'frameshift_deletion', 'frameshift_insertion', 'nonframeshift_deletion', 'nonframeshift_ insertion', 'stopgain', and 'stoploss'; (4) not in dispensable genes as defined by *MacArthur et al., 2012*, and *Rausell et al., 2020*.

IPA was also applied to the WGS data. For detailed methods on IPA please see the Phosphoproteome section of the Materials and methods. For IPA WGS, briefly, gene variations only found in I-ASD, with functional impact, and with a read quality of 'pass', were isolated. Then, for each I-ASD family, variants present only in the affected individual were extracted. The Gene ID for this variant and Gnomad frequency were uploaded to IPA. Core analysis was completed same as for the p-proteome data.

## Immunocytochemistry

Cells were fixed in 4% PFA for 15 min and then permeabilized with 0.3% Triton X-100 in PBS for 10 min. Then, cells were blocked with 10% Normal Goat Serum for 1 hr and incubated in the appropriate

primary antibodies from mouse or rabbit: Sox2 (1:1000, Abcam), Oct4 (1:250, Santa Cruz, Sc-5279), Nestin (1:500, R&D Systems, MAB1259), Pax6 (1:300, Covance, PRB-278P), β-III tubulin (TuJ1, 1:200 Covance, MMS-435P), Tau (1:500, Santa Cruz, Sc-5587), O4 (R&D Systems, MAB1326 1:500), GalC (Santa Cruz 1:250 sc-518055), and Glial fibrillary acidic protein (GFAP, 1:1000, Dako, G9269). Then, cells were washed three times for 5 min with 1× PBS and incubated 1 hr in appropriate secondary antibody at 1:1000. The secondary antibodies used were red (594 Alexa Fluor, Thermo Fisher Scientific, Invitrogen, A-11032, A-11037) or green (488 Alexa Fluor, Thermo Fisher Scientific, Invitrogen, A-11001, A-11070) goat-anti-mouse antibodies or goat-anti-rabbit antibodies. Cells were visualized on fluorescent microscope.

## Media for experimental conditions

The protocols for all experimental media conditions, neurite outgrowth, and neurosphere assays are described in detail in our prior JOVE paper (*Williams et al., 2018*).

All experiments were conducted in 30% neural expansion media (30 NEM) which was made by diluting NEM by 70% in a 1:1 ratio of DMEM/F12 and NB media. Primocin antibiotic (100 µg/mL, InvivoGen ant-pm-1) was also added to the media.

## Neurite outgrowth assay

50,000 NPCs were plated onto a 35 mm dish coated with 0.1 mg/mL PDL and 5 µg/mL Fibronectin (Sigma-Aldrich F1141) containing 1 mL of 30 NEM without or with drugs/EFs. Vehicle, EF, or small molecule inhibitor/activator were added to media prior to cell plating. The EFs and concentrations used include: NGF (Peprotech, AF-450-01) at 3, 10, 30, and 100 ng/mL dissolved into 30 NEM; PACAP (American Peptide/BACHEM, H-8430) at 1, 3, 10, 30, and 100 nM dissolved into 30 NEM; and Serotonin (Sigma-Aldrich, H9523) at 30, 100, 300 µg/mL dissolved into 30 NEM. The small molecule activator and inhibitor and concentrations were: SC79 (AKT activator, S7863) in DMSO diluted into 30 NEM at concentration of 0.1, 0.3, 1, 3, 6 µg/mL and MK-2206 (AKT inhibitor, S1078) in DMSO diluted into 30 NEM at concentrations of 0.3, 1, 3, 10, 30, and 100 nM. Vehicles were matched to whichever carrier was used to dissolve the drug or EF (e.g. for small molecule studies, vehicle was DMSO dissolved into 30 NEM and volumes equivalent to the drug were added to each dish). Drugs and EFs were not replenished, and media was not changed in the 48 hr period of incubation and as such the initial EFs/molecules added at the start of the experiment remained for the full 48 hr period.

For each condition (control or EF or small molecule), two to three dishes were set up in each experiment. NPCs were incubated at 37°C for 48 hr and then fixed in 4% PFA. In each dish, the proportion of cells bearing neurites was counted blind. Neurites were defined as processes that extend from the cell body that are equal to or greater than 2 cell body diameters in length. For cells with more than one process, the longest process was assessed. Cells were counted directly on a phase contrast microscope at 32×. Two to four randomly chosen 1 cm rows were counted per dish. At least 150 cells were counted per dish and the proportion of cells with neurites was calculated in each dish.

## Neurosphere migration assay

Neurospheres were formed by dissociating confluent NPCs and plating 1 million cells into uncoated 35 mm dishes with 100 NEM. No small molecules or EFs were added during this formation step. NPCs were then incubated at 37°C for 24–96 hr (varied from line to line) to allow aggregation into neurospheres. Sphere size was assessed daily using a live-ruler on a phase contrast microscope. When a majority of spheres reached an approximate diameter of 100 µm (±20 µm), the migration assay was performed.

To coat plates, a Matrigel aliquot was dissolved into 6 mL of ice-cold 30 NEM. Appropriate vehicles and EFs were added to the 30 NEM/Matrigel mixture as desired. For migration the following EFs and small molecule inhibitors were used as noted above in the neurite section (PACAP, SC-79, MK-2206). Then, 1 mL of the 30 NEM/Matrix ± EF mixture was added into the six-well plate. Plates were then incubated for 30 min at 37°C. While plates were incubating, neurospheres were collected from the 35 mm dishes and placed into a 15 mL conical tube. The tube was centrifuged at 100×*g* and the pelleted spheres were gently resuspended using a P1000 in 1–3 mL of pre-warmed 30 NEM (with appropriate concentration of EF or small molecule added). 200 µL of the sphere suspension was then

placed into the 30 NEM Matrigel plates. Spheres were allowed to migrate for 48 hr and were fixed for 30 min in 4% PFA, washed, and stored in PBS + 0.05% sodium azide.

Neurospheres were imaged at 10× on a phase contrast microscope. At least 15 neurospheres across all dishes were imaged per condition. Spheres that were not in contact with another exhibited a contiguous migrating cell carpet and intact inner cell mass were imaged. Average migration was measured using ImageJ. To measure migration, the outer contour of a neurosphere was traced using the freehand line tool and the enclosed area was measured. Then, the inner cell mass area was measured. Migration was quantified by subtracting the inner cell mass area from the total neurosphere area. At least 15 neurospheres were analyzed per condition. Exclusion criteria are detailed in JOVE methods paper (*Williams et al., 2018*).

Given that neurospheres are not the same size when plated, it was important to ensure that initial sphere size (ISS) did not affect migration. As such, proof of principal tests were conducted to assess if ISS influenced migration. Migration and ISS data were obtained from each 'diagnosis' group (Sib, NIH, I-ASD, 16pASD) over two experiments with 15 neurospheres each. Linear modeling was conducted in R studio utilizing the basic equation lm(Migration~ISS±diagnosis) where 'lm' allows R to conduct linear modeling, ~ allows for R to calculate the relationship between migration and ISS utilizing any variables and data specified. Results of these analyses and others are found in the supplement, but ultimately, these studies showed us that the ISS did not affect migration, whereas diagnosis did (*Figure 1— figure supplement 1*). Thus, for all remaining neurosphere studies, ISS was not factored into analyses.

## Western blot

To collect protein, confluent NPCs were dissociated, pelleted, and plated onto PDL + Fibronectin-coated plates with 1 mL 30 NEM. 1 million NPCs were plated in 35 mm dishes and incubated for 48 hr at 37°C. At 48 hr, cells were treated with vehicle or drug (SC-79 or MK-2206) for 20 or 30 min. Then, dishes were washed with ice-cold PBS two times. Cells were then lysed by adding 30 µL of ice-cold lysis buffer/dish and scraping with cell scraper. Lysate was then sonicated 2× for 1 min on ice and then samples were centrifuged at 4°C for 10 min. The supernatant was removed and saved while the pellet was discarded. Protein was aliquoted and stored at –80°C.

For small molecule drugs utilized (SC-79 and MK-2206), studies in other cellular systems had shown appropriate increase and reduction in p-S6, respectively, with drug (*Hao et al., 2016*; *Zhu et al., 2019*; *Chen et al., 2017*; *Simioni et al., 2013*; *Somnay et al., 2013*; *Wilson et al., 2014*; *Luan et al., 2018*). However, at the time of our studies, there had been no published papers showing the effects of these drugs on human NPCs. As such, our initial studies with these molecules involved confirming the effect of SC-79 and MK-2206 on p-AKT and p-S6 levels in our NPCs. We initially focused our studies on two I-ASD pairs, Family-1 in which I-ASD NPCs had reduced p-S6 levels and Family-2 in which I-ASD NPCs had increased p-S6 levels. We trialed multiple concentrations (SC-79: 0.1, 0.3, 1, 2, 3, 10 µg/mL, MK-2206, 0.3, 1 3, 10, 30, 100 nM) of each small molecule drug as well as multiple time points (5 min, 10 min, 20 min, 30 min, 1 hr, 24 hr, and 48 hr) to arrive at the doses and 20 and 30 min stimulation paradigm applied in this manuscript (data not shown). Given thorough testing in these two families we were able to clearly establish that SC-79 increases both p-AKT and p-S6 and MK-2206 decreases both p-AKT and p-S6 in Sib-1, I-ASD-1, Sib-2, and I-ASD-2. As such, in the other NPCs (Family-3, 16pDel and NIH), we did not conduct rigorous tests with these small molecules.

Aliquoted samples were thawed and 20 µg of protein was mixed with appropriate volumes of 4× NuPAGE LDS Sample buffer and 10× NuPAGE Sample Reducing Agent. All immunoblotting reagents were obtained from Thermo Fisher Scientific. Samples were boiled, cooled, and loaded into wells on a 12% SDS-PAGE poly-acrylamide gel. The samples were run at 100 V for 1.5 hr in an electrophoresis apparatus with 1× NuPAGE Mops SDS Running Buffer. Protein was then transferred to a PVDF membrane using a wet transfer apparatus with 1× NuPAGE Transfer buffer. The membranes were washed with Tris-buffered saline (TBS) with Tween 20 (0.1%) (TBS-T) and blocked with 5% powdered milk in 0.1% TBS-T. Then, membranes were probed for proteins with the following antibodies used at a 1:2000 concentration and incubated overnight at 4°C: S6 (2317, Cell Signaling), p-S6 (2211, Cell Signaling) as previously reported (*Genestine et al., 2015*; *Yan et al., 2013b*, *Mairet-Coello et al., 2009*; *Mairet-Coello et al., 2012*; *Tury et al., 2011*). GADPH was used as a loading control and was probed with antibody (Meridian Life Sciences, H86045M) at a dilution of 1:10,000 for 1 hr. After incubation at 1°C, membrane was washed and appropriate HRP-conjugated 2°C antibody was applied

at a 1:1000 concentration for all antibodies except GAPDH, for which 2°C antibody was at 1:5000. Then, ECL was applied to membranes for 1 min and membrane was applied to medical grade X-ray film. For most experiments, two gels were loaded with identical protein samples and then each gel was transferred to a separate membrane. For experiments requiring fewer samples, a single gel was run and then cut in half prior to transfer. Multiple wells were loaded, often with protein from different clones and different neural inductions. While each of these gels had identical samples loaded, one gel was utilized to probe for all phospho-proteins and the other for total protein to avoid stripping and to decrease background signal. Membranes were often probed with multiple antibodies. Each primary was incubated on its own for 2 hr at room temperature or overnight at 4°C, followed by the appropriate secondary and ECL steps. Film exposure time varied by antibody and often multiple films were exposed to chemiluminescent bands and then the films that were not over or undersaturated were selected for scanning into JPEG. After the above process was completed for one antibody of interest, the film was washed for 1 hr with TBS and then reprobed with another primary antibody. Often, the first antibody utilized was the one that had the weakest signal. For our experiments the order of antibodies was as follows: p-AKT, p-S6, and then GAPDH. Scanned films were then quantified on ImageJ. Band intensities for phospho- and total protein were normalized to signals for GAPDH. The GAPDH-normalized phospho-antibody intensity was divided by the GAPDH-normalized total antibody to get a relative protein intensity.

## Generation of western blot representative images

For generation of representative images, X-ray films were scanned into PDF files. PDF files were then converted to JPEGs which were loaded into Adobe Photoshop 2022. Grayscale was applied for enhanced visibility and equivalent brightness and contrast were applied to all blots used in the same figure on the same scan. In most instances, the rectangular marquee tool was used to select either individual lanes or several lanes (depending on whether originals were appropriate contiguous or not). Then the copy function was used to make a copy of the selected area and lanes were placed together as needed for generation of representative images. In some cases, for figures where lanes were appropriately spaced, the delete function was used to remove lanes not needed in the figure and then remainder of image was cropped to make representative blots. No size alterations were made to individual bands, no brightness contrast function was applied to individual bands, and no other editing was made to individual bands other than cropping. Unedited images of the representative blots have been made available for comparison.

## Phospho-proteome

### Protein collection

As described above, protein was collected from NPCs plated on 0.1 mg/mL PDL+5 µg/mL FN at a density of 1 million in 35 mm dishes for 48 hr. At least 1.5 mg of protein was collected per individual for p-proteomics. On average, 50–150 µg of protein was acquired from one dish of 1 million cells. Thus, to acquire 1.5 mg of protein, samples of protein were pooled from multiple experiments over time. For final samples, protein was collected from three to four experiments (of three to five dishes) across three different passages and two different neural inductions to ensure adequate sample and to account for variability. Samples were frozen at –80°C after each collection and then ultimately thawed, pooled, and sonicated.

### Proteomic analysis

500 µg of each lysate from above were digested using standard FASTP protocol while 450 µg of each sample was labeled with TMT10 plex reagent (Thermo Scientific) according to the manufacturer's instructions and combined after labeling and dried (*Mertins et al., 2013*). TMT labeled and combined sample were desalted with SPEC C18 column and solubilized in 200 µL of buffer A (20 mm ammonium formate, pH 10) and separated on an Xbridge column (Waters; C18; 3.5 µm, 2.1 × 150 mm$^2$) using a linear gradient of 1% B min$^{-1}$, then from 2% to 45% B (20 mm ammonium formate in 90% acetonitrile, pH 10) at a flow rate of 200 µL/min using Agilent HP1100. Fractions were collected at 1 min intervals and dried under vacuum. For total proteome analysis, 14 fractions (fraction from 27 to 40 min) were chosen. 5% of each fraction were desalted with stage tip and analyzed by nano-LC-MS/MS. For p-proteomics, fractions were combined in concatenate style to make six fractions.

IMAC enrichment of phosphopeptides was adapted from **Mertins et al., 2013**, with modifications. Ion-chelated IMAC beads were prepared from Ni-NTA Superflow agarose beads (QIAGEN, MA, USA). Nickel ion was stripped with 50 mM EDTA and iron was chelated by passing the beads through aqueous solution of 200 mM $FeCl_3$ followed by three times of water wash and one time wash with binding buffer (40% acetonitrile, 1% formic acid). Combined HPH RP fractions were solubilized in binding buffer and incubated with IMAC beads for 1 hr. After three times of wash with binding buffer, the phosphopeptides were eluted with 2× beads volume of 500 mM potassium hydrogen phosphate, pH 7.0, and the eluate was neutralized with 10% formic acid. The enriched phosphopeptides were further desalted by Empore 3M C18 (2215) StageTip[2] prior to nanoLC-MS/MS analysis.

Nano-LC-MS/MS was performed using a Dionex rapid separation liquid chromatography system interfaced with a QExactive HF (Thermo Fisher Scientific). Samples were loaded onto an Acclaim PepMap 100 trap column (75 µm × 2 cm, Thermo Fisher) and washed with Buffer A (0.1% trifluoroacetic acid) for 5 min with a flow rate of 5 µL/min. The trap was brought in-line with the nano analytical column (nanoEase, MZ peptide BEH C18, 130A, 1.7 µm, 75 µm×20 cm, Waters) with flow rate of 300 nL/min with a multistep gradient. Mass spectrometry data were acquired using a data-dependent acquisition procedure with a cyclic series of a full scan acquired with a resolution of 120,000 followed by tandem mass spectrometry (MS/MS) scans (30% collision energy in the HCD cell) with resolution of 30,000 of the 20 most intense ions with dynamic exclusion duration of 20 s.

For both proteome and p-proteome studies, Maxquant identification and Perseus for quantitation were employed and an in-house R program that compounds variability at each level of the analysis (spectra to peptides, peptides to proteins, proteins per experimental group) was utilized to determine statistical significance. Specifically, significance for group comparisons used Student's t-tests with equal variance on both sites and q-values will be calculated using a 5% false discovery rate. Volcano plots were generated to illustrate individual peptides and p-peptides that are different between ASD cases and controls. Both statistically significant differences (p-value ≤ 0.05/number of observations after Bonferroni correction) and >2-fold differences peptide amounts will be denoted. As such initial data included over 9700 proteins, however, to appropriately adjust for multiple comparisons we only selected proteins which met the criteria of significance threshold greater than log p of 5 for further analyses. This analysis was performed between: I-ASD vs Sib controls, 16p-Del vs Sib controls, and I-ASD vs 16pDel NPCs.

## G-profiler analysis

Our p-proteomic dataset, which included UniProt Identifiers and comparisons of fold changes relative to control (Sib), were submitted into the system. The specific tool utilized was g: GOSt which performs statistical enrichment analysis to interpret a user-provided gene list. The data provided users multiple sources of functional evidence including Gene Ontology (GO) terms, biological pathways, regulatory motifs of transcription factors and microRNAs, human disease annotations, and protein-protein interactions. For analysis, g:Profiler allows users to set preferences on statistical domain scope, significance threshold, and user threshold. Briefly, for the statistical domain scope, the program was tasked to look through all known genes, allowing for calculation of statistical significance considering all genes of the human genome in the Ensembl database. Next, for the significance threshold the g:SCS algorithm was used, which sets an experiment-wide threshold of p = 0.05. Finally, the user-defined p-value threshold allows for further filtering of results; this threshold was also set to p = 0.05. As a readout of the results, the program looks at GO, which is a bioinformatics initiative that defines biological functions and their relationship to one another. The three major subontologies are molecular functions, gene products functions, biological process, which describes biological processes the gene product participates in, and cellular component, describes in which part of the cell the gene product is physically located.

## Ingenuity Pathway Analysis

Our phospho- and total proteomic dataset, which included UniProt Identifiers, p-values, q-values, and log2 ratios of comparisons of fold changes of identified phosphorylated proteins relative to Sib, were submitted into IPA for core analysis. The pathways and functional networks presented were all generated through the use of IPA QIAGEN Inc, (https://www.qiagenbioinformatics.com/products/ingenuity-pathway-analysis/). The core analysis generated molecular networks according to biological as well as molecular functions including canonical pathways, disease-based functions, and upstream

regulatory analysis for the discovery of strongly differential molecules in our dataset. Both direct and indirect relationship between molecules based on experimentally observed data and data sources were considered in human databases in the Ingenuity Knowledge Base. Right-tailed Fisher's exact test was used to determine the probability that biological functions and/or diseases were over-represented in the protein dataset. Z-scores compiled as a statistical result of differential protein expression according to the fold changes were also compiled.

## Compilation of data and statistical analyses

Data acquired from the idiopathic autism individuals were compared to data acquired from unaffected Sibs. Data was compiled and organized in several ways. Comparisons were made within Family (Sib-1 vs ASD-1), across families (Sib-1 vs ASD-2), or as an average of all Sib (three patients, all clones) compared to the average data from all idiopathic ASD (three patients, all clones). For every neurite experiment, two to three dishes were set up per condition. For every migration condition, 20–30 spheres were evaluated per condition. Dishes were kept as separate N-values. For example, when compiling data for Sib-1 in control condition, all neurite dishes from every clone were averaged together to get a mean neurite value for this individual but SEM was calculated utilizing each dish as a separate 'N' value. For spheres, each neurosphere was considered a separate statistical point. Migration values from each sphere derived from multiple clones were averaged together to acquire an average migration value for each patient. Spheres were also analyzed by clone. For the 16p11.2, data was compared to composite averages of the Sibs (dishes from all three Sib patients and all clones) or averages of the NIMH controls (dishes from both patients, all neural inductions). Assuming statistical normality, in comparisons that include only two groups, unpaired t-test was used to test differences in affected vs unaffected samples. Simple unpaired or paired t-tests were run in Microsoft Excel. If normality assumption was not satisfied, the non-parametric Wilcoxon test was used in GraphPad Prism 8. The presence of multiple ASD subtypes and the use of EFs lead to instances where multiple group comparison is necessary. For such cases, in normal data, analysis of variance or ANOVA (one or two way) was used to detect statistical significance. To reduce type I error, p-values for the ANOVAs were calculated using Tukey correction in GraphPad Prism 8. If normality assumption did not hold, non-parametric Kruskal-Wallis test was applied for multiple comparisons.

## Acknowledgements

Our sincerest thanks to the NJLAGs families whose interest in supporting autism research allowed for the findings in our studies and Drs. Linda Brzustowicz and Judy Flax for their collaboration and support. We also give our sincerest thanks to the families participating in the Simons Searchlight Consortium. We appreciate obtaining access to the phenotypic data and the cellular biospecimens on the SFARI base as well. We thank RUCDR Infinite Biologics for providing Simons Searchlight 16p11.2 Deletion and NIH Regenerative Medicine Program iPSCs as well as for their support and technical guidance throughout the study. Our thanks to the Center for Integrative Proteomics Research and the Mass Spectrometry Facility of Rutgers Robert Wood Johnson Medical School for their support and technical guidance for the proteomic analyses. We would like to thank the following individuals for their contributions to this manuscript: Courtney McDermott, Maya Hale, Katelyn Jo, Angelique Sarrosquy, and Meghan Eller.

We would like to acknowledge the following funding sources as well. This work was supported by the New Jersey Governor's Council for Medical Research and Treatment of Autism (CAUT13APS010, CAUT14APL031, CAUT15APL041, CAUT19APL014) and the Nancy Lurie Marks Family Foundation for ED-B and JHM; the NJ Health Foundation (PC 63-19) for JHM; the Mindworks Charitable Lead Trust and Jewish Community Foundation of Greater MetroWest for ED-B; an Autism Science Foundation Undergraduate summer research grant for CP and ED-B and Rutgers School of Graduate Studies for SP and ED-B. JX was supported by the New Jersey Governor's Council for Medical Research and Treatment of Autism grant CAUT19APL028. Lastly, the NIHR25 'Educating Physician Scientist in Psychiatry' grant (5R25MH119043-05) for SP.

## Additional information

### Funding

| Funder | Grant reference number | Author |
|---|---|---|
| New Jersey Governor's Council for Medical Research and Treatment of Autism | CAUT13APS010 | James H Millonig Emanuel DiCicco-Bloom |
| New Jersey Governor's Council for Medical Research and Treatment of Autism | CAUT14APL031 | Emanuel DiCicco-Bloom James H Millonig |
| New Jersey Governor's Council for Medical Research and Treatment of Autism | CAUT15APL041 | James H Millonig Emanuel DiCicco-Bloom |
| New Jersey Governor's Council for Medical Research and Treatment of Autism | CAUT19APL014 | James H Millonig Emanuel DiCicco-Bloom |
| Nancy Lurie Marks Family Foundation | | Emanuel DiCicco-Bloom James H Millonig |
| New Jersey Health Foundation | PC 63-19 | James H Millonig |
| Mindworks Charitable Lead Trust | | Emanuel DiCicco-Bloom |
| Jewish Community Foundation of Greater MetroWest NJ | | Emanuel DiCicco-Bloom |
| Autism Science Foundation | Summer Undergraduate Research Grant | Cynthia Peng |
| Rutgers School of Graduate Studies | Thesis Finishing Grant | Smrithi Prem |
| New Jersey Governor's Council for Medical Research and Treatment of Autism | CAUT19APL028 | Smrithi Prem Jinchuan Xing Emanuel DiCicco-Bloom |
| National Institutes of Health | R25 5R25MH119043-05 | Smrithi Prem |

The funders had no role in study design, data collection and interpretation, or the decision to submit the work for publication.

### Author contributions

Smrithi Prem, Conceptualization, Data curation, Formal analysis, Supervision, Validation, Investigation, Visualization, Methodology, Writing – original draft, Writing – review and editing; Bharati Dev, Data curation, Formal analysis, Investigation, Visualization; Cynthia Peng, Data curation, Formal analysis, Visualization; Monal Mehta, Resources, Validation, Methodology; Rohan Alibutud, Resources, Formal analysis; Robert J Connacher, Resources, Formal analysis, Investigation, Visualization, Methodology, Writing – review and editing; Madeline St Thomas, Conceptualization, Resources, Formal analysis; Xiaofeng Zhou, Data curation, Formal analysis, Investigation, Project administration; Paul Matteson, Conceptualization, Resources, Data curation, Methodology; Jinchuan Xing, Conceptualization, Resources, Software, Supervision, Funding acquisition, Investigation, Methodology; James H Millonig, Conceptualization, Resources, Supervision, Funding acquisition, Validation, Investigation, Visualization, Methodology, Writing – original draft, Writing – review and editing; Emanuel DiCicco-Bloom, Conceptualization, Resources, Data curation, Formal analysis, Supervision, Funding acquisition, Validation, Investigation, Visualization, Methodology, Writing – original draft, Writing – review and editing

## Author ORCIDs

Smrithi Prem http://orcid.org/0000-0001-6961-7904
Jinchuan Xing http://orcid.org/0000-0001-6469-8733
Emanuel DiCicco-Bloom https://orcid.org/0000-0001-5091-1046

## Decision letter and Author response

Decision letter https://doi.org/10.7554/eLife.82809.sa1
Author response https://doi.org/10.7554/eLife.82809.sa2

## Additional files

### Supplementary files

• Supplementary file 1. Tabulation of neural precursor cell (NPC) N-values for *Figures 1–7*. In each cell, multiple different kinds of N-values are represented: # of clones (C)/Total # of neural inductions (NI)/# of experiments (E) and for neurite experiments, # of dishes (D) whereas for neurospheres experiment, # of neurospheres (NS).

• Supplementary file 2. Alternative allele genotypes in the chr16.p11.2 deletion region in three I-ASD families. The count of the heterozygous and homozygous alternative allele genotypes and their ratio in the chr16.p11.2 deletion region (chr16: 28,500,001–35,300,000). In the event of a chr16.p11.2 deletion, we expect no heterozygous genotypes in the region. The large number of heterozygous genotypes across all individuals in this region indicates that the deletion does not appear in any of the individuals in the three I-ASD families.

• MDAR checklist

### Data availability

Genome Wide Sequencing data has been deposited into the NIH NDA. All excel sheets for graphs in the manuscript as well as unedited western blot films (labeled and unlabeled) have been deposited in Dryad: https://doi.org/10.5061/dryad.6wwpzgn5v.

The following dataset was generated:

| Author(s) | Year | Dataset title | Dataset URL | Database and Identifier |
|---|---|---|---|---|
| Prem S | 2024 | Data for: Dysregulation of mTOR signaling mediates common neurite and migration defects in both idiopathic and 16p11.2 deletion autism neural precursor cells | https://doi.org/10.5061/dryad.6wwpzgn5v | Dryad Digital Repository, 10.5061/dryad.6wwpzgn5v |

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

## Appendix 1

**Appendix 1—key resources table**

| Reagent type (species) or resource | Designation | Source or reference | Identifiers | Additional information |
|---|---|---|---|---|
| Antibody | Phospho-S6 Ribosomal Protein (Ser235/236) Antibody Rabbit polyclonal | Cell Signaling Technology | Catalog #: 2211 | 'p-S6' WB: (1:2000) |
| Antibody | S6 Ribosomal Protein (54D2) Mouse monoclonal | Cell Signaling Technology | Catalog #: 2317 | 'Total S6 or S6' WB: (1:2000) |
| Antibody | Phospho-Akt (Ser473) Antibody Rabbit polyclonal | Cell Signaling Technology | Catalog #: 9271 | 'p-AKT' WB: (1:1000) |
| Antibody | Akt Antibody Rabbit polyclonal | Cell Signaling Technology | Catalog #: 9272 | 'AKT' WB: (1:2000) |
| Antibody | Glyceraldehyde-3-phosphate dehydrogenase Mouse Monoclonal Antibody, Unconjugated, Clone 4G5 | Meridian Life Science | Catalog #: H86045M | 'GAPDH' WB: (1:10,000) |
| Antibody | Goat anti-Rabbit IgG (H+L) Secondary Antibody, HRP | Thermo Fisher Scientific | Catalog #: 31460 | WB: (1:2000) WB GAPDH: (1:5000) |
| Antibody | Goat anti-Mouse IgG (H+L) Secondary Antibody, HRP | Thermo Fisher Scientific | Catalog #: 31430 | WB: (1:2000) WB GAPDH: (1:5000) |
| Antibody | Recombinant Anti-SOX-2 antibody Rabbit monoclonal | Abcam | Catalog #: Ab92494 | 'SOX2 antibody' WB: (1:4000) ICC: (1:1000) |
| Antibody | Human Nestin Antibody Mouse Monoclonal | R&D Systems | Catalog #: MAB1259 | 'Nestin' WB: (1:5000) ICC: (1:500) |
| Antibody | Purified anti-PAX-6 antibody Rabbit Polyclonal | BioLegend (Previously Covance) | Covance Catalog #: PRB-278P | 'Pax-6' WB: (1:2000) ICC: (1:500) |
| Antibody | Purified anti-tubulin β 3 (TUBB3) Antibody Mouse Monoclonal | BioLegend (Previously Covance) | Covance Catalog #: MMS-435P | 'TUJ1' WB: (1:1000) ICC: (1:100) |
| Antibody | Goat anti-mouse IgG (H+L) Cross-Adsorbed Secondary Antibody, Alexa Fluor 488 | Thermo Fisher Scientific | Catalog #: A-11001 | ICC: (1:2000) |
| Antibody | F(ab')2-Goat anti-rabbit IgG (H+L) Cross-adsorbed secondary antibody, Alexa Fluor 488 | Thermo Fisher Scientific | Catalog #: A-11070 | ICC: (1:2000) |
| Antibody | Goat anti-Mouse IgG (H+L) Highly Cross-Adsorbed Secondary Antibody, Alexa Fluor 594 | Thermo Fisher Scientific | Catalog #: A-11032 | ICC: (1:2000) |
| Antibody | Goat anti-Rabbit IgG (H+L) Highly Cross-Adsorbed Secondary Antibody, Alexa Fluor 594 | Thermo Fisher Scientific | Catalog #: A-11037 | ICC: (1:2000) |
| Cell Line (Human) | NCRM-1 | NIH | | Research Grade: RMP Generated iPSC Line, control reference line, CD34+ Cord blood, episomal plasmid reprogramming |

*Appendix 1 Continued on next page*

*Appendix 1 Continued*

| Reagent type (species) or resource | Designation | Source or reference | Identifiers | Additional information |
|---|---|---|---|---|
| Cell Line (Human) | NCRM-3 | NIH | | Research Grade: RMP Generated iPSC Line, control reference line, CD34+ Cord blood, episomal plasmid reprogramming |
| Cell Line (Human) | NCRM-6 | NIH | | Research Grade: RMP Generated iPSC Line, control reference line, CD34+ Cord blood, episomal plasmid reprogramming |
| Cell Line (Human) | iPSC Lines (16p11.2 deletion) | This paper | | 2 male and 1 female line. One clone each. Maintained by Millonig lab |
| Cell Line (Human) | iPSC Lines (I-ASD) | This paper | | 3 male lines, 3 clones each Maintained by Millonig lab |
| Cell Line (Human) | IPSC Lines (Sib) | This paper | | 3 male lines, 3 clones each. Maintained by Millonig lab |
| Chemical Compound, drug | Primocin | Invivogen | Catalog #: ant-pm-1 | Antimicrobial for culture Concentration: 100 µg/mL |
| Chemical Compound, drug | Serotonin Hydrochloride | Sigma-Aldrich | Catalog #: H9523 | 'Serotonin or 5-HT' |
| Chemical Compound, drug | Y-27632 | STEMCELL Technologies | Catalog #: 72302 | Noted as 'Y- compound' in manuscript, ROCK inhibitor |
| Chemical Compound, drug | SC79 | Selleckchem | Catalog #: S7863 | Small molecule drug, AKT activator |
| Chemical Compound, drug | MK-2206 2HCl | Selleckchem | Catalog #: S107 | Small molecule drug, AKT inhibitor |
| Commercial assay, kit | MycoAlert Mycoplasma Detection Kit | Lonza | Catalog #: LT07-318 | |
| Commercial assay, kit | mTeSR1 Complete Kit | STEMCELL Technologies | Catalog #: 5850 | Includes - mTeSR Basal medium (400 mL) - mTeSR 5× supplement 100 mL |
| Commercial assay, kit | Gibco PSC Neural Induction Medium | Thermo Fisher Scientific | Catalog #: A1647801 | Includes: - Basal medium (500 mL) - Supplement (10 mL) |
| Other | Gibco DMEM/F12 | Thermo Fisher Scientific | Catalog #: 11320033 | Cell Culture Media |
| Other | Gibco Neurobasal Medium | Thermo Fisher Scientific | Catalog #: 21103049 | Cell Culture Media |
| Other | Costar 6-well clear TC treated multiple well plates | Corning LifeSciences | Product #: 3506 | Cultureware; six-well plates |
| Other | 35 mm TC-treated culture dish | Corning | Product #: 430165 | Cultureware |
| Other | Accutase | Thermo Fisher Scientific | Catalog #: A111050 | Cell detachment solution |
| Other | Matrigel Matrix | Corning LifeSciences | Product #: 354277 | Extracellular Matrices |

*Appendix 1 Continued on next page*

*Appendix 1 Continued*

| Reagent type (species) or resource | Designation | Source or reference | Identifiers | Additional information |
|---|---|---|---|---|
| Other | Fibronectin bovine plasma | Sigma-Aldrich | Catalog # F1141 | Sterile filtered Extracellular Matrices |
| Other | NuPAGE LDS Sample Buffer (4×) | Thermo Fisher Scientific | Catalog #: NP0007 | Gel Electrophoresis Equipment and Supplies |
| Other | NuPAGE Sample Reducing Agent (10×) | Thermo Fisher Scientific | Catalog #: NP0004 | Gel Electrophoresis Equipment and Supplies |
| Other | NuPAGE 12%, Bis-Tris, 1.0 mm, Mini Protein Gels | Thermo Fisher Scientific | Catalog #: NP0342PK2 | Gel Electrophoresis Equipment and Supplies |
| Other | NuPAGE MOPS SDS Running Buffer (20×) | Thermo Fisher Scientific | Catalog #: NP000102 | Gel Electrophoresis Equipment and Supplies |
| Other | PVDF Transfer Membranes, 0.45 µm | Thermo Fisher Scientific | Catalog #: 88585 | Gel Electrophoresis Equipment and Supplies |
| Other | NuPAGE Transfer Buffer (20×) | Thermo Fisher Scientific | Catalog #: NP00061 | Gel Electrophoresis Equipment and Supplies |
| Other | Pierce ECL Western Blotting Substrate | Thermo Fisher Scientific | Catalog #: 32106 | Gel Electrophoresis Equipment and Supplies |
| Other | Pierce ECL Western Blotting Substrate | Thermo Fisher Scientific | Catalog #: 32106 | Gel Electrophoresis Equipment and Supplies |
| Other | LucentBlue X-Ray Film | Advantsa | Catalog #: 1190V51 or EK-5150 | Gel Electrophoresis Equipment and Supplies |
| Peptide, recombinant protein | Recombinant human β-NGF | Peprotech | Catalog #: 450-01 | 'Nerve Growth Factor or NGF' |
| Peptide, recombinant protein | PACAP-38 (human, mouse, ovine, porcine, rate) | BACHEM | Product #: 4031157 Previous Product #: H-8430 | 'Pituitary adenylate cyclase activating polypeptide or PACAP' |
| Software | Ingenuity Pathway Analysis | QIAGEN | | p-Proteome, proteome, and WGS pathway analysis |
| Software | Photoshop 2023 | Adobe | | Image Editing |
| Software | ImageJ | NIH | | Western blot quantification |
| Software | Prism | GraphPad by Dotmatics | | Statistical analysis, graph generation |
| Software | RStudio | GNU Project | | Modeling, statistical analysis |

