## [Editor Report]

This important study identifies common developmental defects in neural precursor cells derived from idiopathic and 16p11.2 deletion ASD individuals. Convincing evidence is presented that mTOR signaling defects are a shared pathogenic mechanism underlying the cellular defects in these genetically distinct ASD subtypes. This work will be of interest to researchers studying neurodevelopment and related disorders.

---

## [Decision Letter]

**Decision letter after peer review:**

Thank you for submitting your article "Dysregulation of mTOR Signaling Mediates Common Neurite and Migration Defects in Both Idiopathic and 16p11.2 Deletion Autism Neural Precursor Cells" for consideration by *eLife*. Your article has been reviewed by 3 peer reviewers, including Genevieve Konopka as the Reviewing Editor and Reviewer #1, and the evaluation has been overseen by Murim Choi as the Senior Editor.

Essential revisions:

1) Please provide uncropped western blot images.

2) Please provide all data points in figures.

3) Please carry out pS6 western blots for all 8 ASD lines.

4) Please cite the paper on Timothy Syndrome (Birey et al. 2022, Cell Stem Cell).

*Reviewer #1 (Recommendations for the authors):*

1) The presentation of the western blot data is a bit unusual with every band cropped out separately. It is not clear whether the samples were run multiple times per antibody or blots were cut and imaged separately.

2) I would caution the use of the term "autism deficits" and "autism phenotypes" as used in the first paragraph of Discussion. Cell lines can not have autism.

*Reviewer #2 (Recommendations for the authors):*

Individually cropped bands are shown as representative western blots (Figure 4-6). Authors should represent crops that include all samples that are compared and provide uncropped raw westerns in supplementary data.

Even though most plots show individual data points, a few plots do not Figure 5,6I-J. Was there a particular reason for this?

*Reviewer #3 (Recommendations for the authors):*

There are several issues with the figures:

In Figure 2A, the data for I-ASD-1 control and PACAP appear to be identical data sets.

In Figure 1, and others, it is sometimes unclear which group comparisons are shown as statistically significant.

In most of the figures, the resolution of the graphs is inconsistent across different panels within the same figure (e.g. Figure 6 C and D vs G and H).

In many of the figures, some panels are partially obscured cutting off figure labels and/or legends.

Scatter plots in several figures are too crowded with the display of individual points, obscuring the comparison of the means. Scatter plots could be presented as supplemental figures.

In Figure 5F, the concentration of MK-2206 is reported as 10 nm, but should it be 30 nM, consistent with Figure 5G and 5H?

---

## [Author Response]

Essential revisions:1) Please provide uncropped western blot images.

We apologize for not providing the uncropped western blot images with the original manuscript. Our western process was generally as follows: gels were loaded with protein of interest and then transferred to a PVDF membrane. We loaded the same protein samples onto two separate gels: Gel 1 would be transferred to a PVDF membrane that would be probed with antibodies against phospho-protein (i.e. P-S6) while Gel 2 would be transferred to a PVDF membrane that would be probed with antibodies against total proteins (ie S6). For each membrane, multiple antibody probes were often used without any stripping (For example, membrane 1 would be probed sequentially for P-S6, P-AKT, and GAPDH). Once antibody incubations were completed, ECL solution was utilized so that the chemiluminescent signal could be detected utilizing X-ray films. Multiple film exposures were conducted per membrane to produce appropriately saturated bands per antibody of interest. For example, we would develop one film for P-S6 and another film for GAPDH. For representative images and quantification, films were then scanned into JPEG images. From these JPEGs, as can be seen, there are often multiple lanes with multiple clones and neural inductions loaded. As such, to make representative images, especially in cases where contiguous bands were not present, we utilized the cropping function to cut out the lanes of interest used these separate lanes to construct the figures. We have added a section on generation of these representative images in our methods section as well. We will also include all PDFs of unedited Western Blots in Dryad with and without labels and descriptions of displayed lanes.

2) Please provide all data points in figures.

With the exception of Figure 3A &D as well as Figure 7 we have restructured all our graphs (specifically in Figures 5&6) to now include each individual data point to match with all our other figures. What each individual point represents is clearly delineated in our supplemental table (Table 1). For Figure 3A&D, proteome canonical pathway enrichments did not have multiple replicates and as such were not made as a scatter. For figure 7 we attempted to convert to a scatter model, however, given the complexity of the figure, we felt the circles could not be adequately seen and they made the figure excessively busy. See Author response image 1 for a sample:

**Author response image 1. sa2fig1:** 

3) Please carry out pS6 western blots for all 8 ASD lines.

We agree that analysis of the other 5 idiopathic autism lines would provide interesting insight on the generalizability of mTOR dysregulation in ASD and help us understand if the bimodal perturbation of mTOR is present in our other idiopathic patients. However, the focus of the current paper is on the surprising discovery of mTOR dysregulation in both idiopathic and 16p11.2 deletion NPCs despite limited genetic overlap. Moreover, we found that regardless of genotype, these mTOR changes are mechanistically linked to the common developmental phenotypes we noted in these individuals as well. We are greatly interested in the PS6 and mTOR phenotype of the remaining idiopathic cohort. Indeed with our collaborator/coauthor Dr. James Millonig we will next study the presence of neurite, migration, and mTOR dysregulation as a feature of ASD, not only in the remaining 5 idiopathic lines, but also in genetically defined syndromes including FXS, TSC, 16p11.2 duplication, and 22q11.2 deletion and compare those with ASD to those without ASD, research now supported by a recent Department of Defense IDA grant funded in 2022. The NPCs currently studied were randomly selected from the larger cohort of 8 and all of these individuals had cryopreserved T-lymphocytes. Multiple iPSC clones were generated and selected for each of the 16 individuals (ASD and Sibs) and then cryopreserved. For the 3 families we selected in our studies, iPSC clones were thawed, passaged 10 times in order to ensure loss of Sendai virus and attain stability, and expanded sufficiently to obtain adequate cells for our studies. Furthermore, NPC induction, which often took several iterations to get healthy NPCs, was performed multiple times for each clone. The NPCs then also needed to be passaged and expanded, particularly for western studies to generate adequate protein to run multiple western studies. This task would likely take a year or more and given that it does not fully adhere to the scope of our paper, we will address this important next step in future work funded by the new grant.

4) Please cite the paper on Timothy Syndrome (Birey et al. 2022, Cell Stem Cell).

We have included this citation as an example of how bimodal perturbations in signaling pathways can converge upon common deficits in cellular migration. We appreciate the suggestion

Reviewer #1 (Recommendations for the authors):1) The presentation of the western blot data is a bit unusual with every band cropped out separately. It is not clear whether the samples were run multiple times per antibody or blots were cut and imaged separately.

We apologize for the confusing presentation of the western blots in this manuscript. The separate cropping was due to how we ran our experiments and selected the lanes for presentation in the manuscript. For example, in some blots we ran several clones, or several neural inductions together and then selected the most “representative” lanes for the figures, leading to the separately cropped lanes displayed. Of course, all bands for all groups that were analyzed in multiple experiments were included for statistical analyses. To address this concern, we have re-made all the western figures such that more of the background of the blots is visible leading to a more seamless and less “disjointed” crop; we also selected lanes that were closer in proximity to each other while preserving the message we were trying to represent. Furthermore, we now also provide all the raw images of the westerns along with a PDF document “key” describing each image. Lastly, we apologize for not clearly describing how we ran and processed our western samples, which has now been addressed above. We have edited the Materials and methods section to more clearly describe this process.

2) I would caution the use of the term "autism deficits" and "autism phenotypes" as used in the first paragraph of Discussion. Cell lines can not have autism.

Apologies for the imprecise terminology! We have edited the manuscript to state “deficits associated with ASD NPCs” or “Autism NPC phenotypes.

Reviewer #2 (Recommendations for the authors):Individually cropped bands are shown as representative western blots (Figure 4-6). Authors should represent crops that include all samples that are compared and provide uncropped raw westerns in supplementary data.

Please see response above for Reviewer 1, Comment (1). We appreciate this reviewer comment ensuring rigor of data acquisition and analysis. We presume the reviewer expects to see, for example, multiples lanes of each group sample, such as I-ASD-1, in immediate sequence, so that one can compare 3 – 5 lanes of I-ASD-1 samples together to look for intragroup variability, and then compare to another similar group of samples. However, because different combinations of multiple families, clones, and neural inductions were performed in parallel over time, we performed westerns in ongoing fashion, rather than storing samples for years and processing them together in a single huge western. This was done so we could know possible outcomes and plan future clonal productions and neural inductions over time. As can be seen in the newly provided unprocessed raw westerns, blots included a mixture of experiments using small groups of different samples (Sib; ASD; NIH; 16p) that would make their complete display confusing to appreciate. But as requested, we have re-made all the western figures and provided all unedited blot images which do give some sense of replicate data.

Even though most plots show individual data points, a few plots do not Figure 5,6I-J. Was there a particular reason for this?

Apologies for not showing all the data points in the later figures. We thought the column graphs would be easier to interpret in some figures which is why we selected them; we have a majority of the graphs such that all data points are shown.

Reviewer #3 (Recommendations for the authors):There are several issues with the figures:In Figure 2A, the data for I-ASD-1 control and PACAP appear to be identical data sets.

We apologize for this oversight! When copying in data from excel to Graphpad the same data-set was copied into the control and PACAP columns. This has been rectified in the revised manuscript

In Figure 1, and others, it is sometimes unclear which group comparisons are shown as statistically significant.

We have re-made the figures such that the comparison data is more easily visualized.

In most of the figures, the resolution of the graphs is inconsistent across different panels within the same figure (e.g. Figure 6 C and D vs G and H).

We have remade all the figures with high resolution images.

In many of the figures, some panels are partially obscured cutting off figure labels and/or legends.

The original PDF submitted did not have the formatting errors, however following website MS construction these errors appeared. We have edited everything and ensured on recheck that no formatting errors are present!

Scatter plots in several figures are too crowded with the display of individual points, obscuring the comparison of the means. Scatter plots could be presented as supplemental figures.

The editor has requested that all our graphs be made into scatter plots as this allows for more transparency for the data. However, we do agree that the scatter sometimes obscures the mean. To address this, we have made the dots in the plots smaller, with thinner borders and made the column and SEM bar thicker for better visibility.

In Figure 5F, the concentration of MK-2206 is reported as 10 nm, but should it be 30 nM, consistent with Figure 5G and 5H?

Thank you for catching this, it should be 30nM!